# Weighted Bellman Backups for Improved Signal-to-Noise in Q-Updates

## Abstract

Off-policy deep reinforcement learning (RL) has been successful in a range of challenging domains. However, standard off-policy RL algorithms can suffer from low signal and even instability in Q-learning because target values are derived from current Q-estimates, which are often noisy. To mitigate the issue, we propose ensemble-based weighted Bellman backups, which re-weight target Q-values based on uncertainty estimates from a Q-ensemble. We empirically observe that the proposed method stabilizes and improves learning on both continuous and discrete control benchmarks. We also specifically investigate the signal-to-noise aspect by studying environments with noisy rewards, and find that weighted Bellman backups significantly outperform standard Bellman backups. Furthermore, since our weighted Bellman backups rely on maintaining an ensemble, we investigate how weighted Bellman backups interact with UCB Exploration. By enforcing the diversity between agents using Bootstrap, we show that these different ideas are largely orthogonal and can be fruitfully integrated, together further improving the performance of existing off-policy RL algorithms, such as Soft Actor-Critic and Rainbow DQN, for both continuous and discrete control tasks on both low-dimensional and high-dimensional environments.

## 1 Introduction

Model-free reinforcement learning (RL), with high-capacity function approximators, such as deep neural networks (DNNs), has been used to solve a variety of sequential decision-making problems, including board games (Silver et al., 2017; 2018), video games (Mnih et al., 2015; Vinyals et al., 2019), and robotic manipulation (Kalashnikov et al., 2018). It has been well established that the above successes are highly sample *inefficient* (Kaiser et al., 2020). Recently, a lot of progress has been made in more sample-efficient model-free RL algorithms through improvements in off-policy learning both in discrete and continuous domains (Fujimoto et al., 2018; Haarnoja et al., 2018; Hessel et al., 2018; Amos et al., 2020). However, standard off-policy RL algorithms can suffer from instability in Q-learning due to error propagation in the Bellman backup, i.e., the errors induced in the target value can lead to an increase in overall error in the Q-function (Kumar et al., 2019; 2020).

One way to address the error propagation issue is to use ensemble methods, which combine multiple models of the value function (Hasselt, 2010; Van Hasselt et al., 2016; Fujimoto et al., 2018). For discrete control tasks, double Q-learning (Hasselt, 2010; Van Hasselt et al., 2016) addressed the value overestimation by maintaining two independent estimators of the action values and later extended to continuous control tasks in TD3 (Fujimoto et al., 2018). While most prior work has improved the stability by taking the minimum over Q-functions, this also needlessly loses signal, and we propose an alternative way that utilizes ensembles to estimate uncertainty and provide more stable backups.

In this paper, we propose ensemble-based weighted Bellman backups that can be applied to most modern off-policy RL algorithms, such as Q-learning and actor-critic algorithms. Our main idea is to reweight sample transitions based on uncertainty estimates from a Q-ensemble. Because prediction errors can be characterized by uncertainty estimates from ensembles (i.e., variance of predictions) as shown in Figure 1(b), we find that the proposed method significantly improves the signal-to-noise in the Q-updates and stabilizes the learning process. Finally, we present a unified framework, coined SUNRISE, that combines our weighted Bellman backups with an inference method that selects actions using highest upper-confidence bounds (UCB) for efficient exploration (Chen et al., 2017).

We find that these different ideas can be fruitfully integrated, and they are largely complementary (see Figure 1(a)).

We demonstrate the effectiveness of the proposed method using Soft Actor-Critic (SAC; Haarnoja et al. 2018) for continuous control benchmarks (specifically, OpenAI Gym (Brockman et al., 2016) and DeepMind Control Suite (Tassa et al., 2018)) and Rainbow DQN (Hessel et al., 2018) for discrete control benchmarks (specifically, Atari games (Bellemare et al., 2013)). In our experiments, SUNRISE consistently improves the performance of existing off-policy RL methods. Furthermore, we find that the proposed weighted Bellman backups yield improvements in environments with noisy reward, which have a low signal-to-noise ratio.

## 2   RELATED WORK

**Off-policy RL algorithms**. Recently, various off-policy RL algorithms have provided large gains in sample-efficiency by reusing past experiences (Fujimoto et al., 2018; Haarnoja et al., 2018; Hessel et al., 2018). Rainbow DQN (Hessel et al., 2018) achieved state-of-the-art performance on the Atari games (Bellemare et al., 2013) by combining several techniques, such as double Q-learning (Van Hasselt et al., 2016) and distributional DQN (Bellemare et al., 2017). For continuous control tasks, SAC (Haarnoja et al., 2018) achieved state-of-the-art sample-efficiency results by incorporating the maximum entropy framework. Our ensemble method brings orthogonal benefits and is complementary and compatible with these existing state-of-the-art algorithms.

**Stabilizing Q-learning**. It has been empirically observed that instability in Q-learning can be caused by applying the Bellman backup on the learned value function (Hasselt, 2010; Van Hasselt et al., 2016; Fujimoto et al., 2018; Song et al., 2019; Kim et al., 2019; Kumar et al., 2019; 2020). By following the principle of double Q-learning (Hasselt, 2010; Van Hasselt et al., 2016), twin-Q trick (Fujimoto et al., 2018) was proposed to handle the overestimation of value functions for continuous control tasks. Song et al. (2019) and Kim et al. (2019) proposed to replace the max operator with Softmax and Mellowmax, respectively, to reduce the overestimation error. Recently, Kumar et al. (2020) handled the error propagation issue by reweighting the Bellman backup based on cumulative Bellman errors. However, our method is different in that we propose an alternative way that also utilizes ensembles to estimate uncertainty and provide more stable, higher-signal-to-noise backups.

**Ensemble methods in RL**. Ensemble methods have been studied for different purposes in RL (Wiering & Van Hasselt, 2008; Osband et al., 2016a; Anschel et al., 2017; Agarwal et al., 2020; Lan et al., 2020). Chua et al. (2018) showed that modeling errors in model-based RL can be reduced using an ensemble of dynamics models, and Kurutach et al. (2018) accelerated policy learning by generating imagined experiences from the ensemble of dynamics models. For efficient exploration, Osband et al. (2016a) and Chen et al. (2017) also leveraged the ensemble of Q-functions. However, most prior works have studied the various axes of improvements from ensemble methods in isolation, while we propose a unified framework that handles various issues in off-policy RL algorithms.

**Exploration in RL**. To balance exploration and exploitation, several methods, such as the maximum entropy frameworks (Ziebart, 2010; Haarnoja et al., 2018), exploration bonus rewards (Bellemare et al., 2016; Houthooft et al., 2016; Pathak et al., 2017; Choi et al., 2019) and randomization (Osband et al., 2016a;b), have been proposed. Despite the success of these exploration methods, a potential drawback is that agents can focus on irrelevant aspects of the environment because these methods do not depend on the rewards. To handle this issue, Chen et al. (2017) proposed an exploration strategy that considers both best estimates (i.e., mean) and uncertainty (i.e., variance) of Q-functions for discrete control tasks. We further extend this strategy to continuous control tasks and show that it can be combined with other techniques.

## 3   BACKGROUND

**Reinforcement learning**. We consider a standard RL framework where an agent interacts with an environment in discrete time. Formally, at each timestep $t$, the agent receives a state $s_t$ from the environment and chooses an action $a_t$ based on its policy $\pi$. The environment returns a reward $r_t$ and the agent transitions to the next state $s_{t+1}$. The return $R_t = \sum_{k=0}^{\infty} \gamma^k r_{t+k}$ is the total accumulated rewards from timestep $t$ with a discount factor $\gamma \in [0, 1)$. RL then maximizes the expected return.

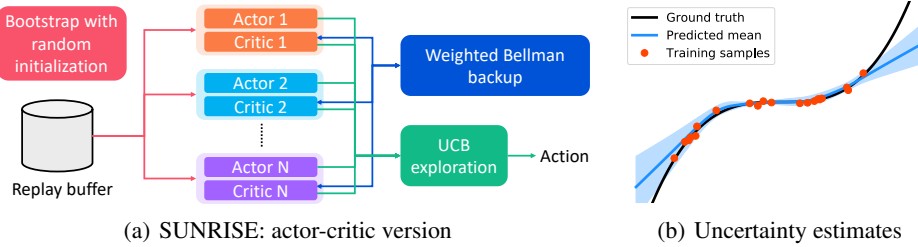

(a) SUNRISE: actor-critic version    (b) Uncertainty estimates

Figure 1: (a) Illustration of our framework. We consider $N$ independent agents (i.e., no shared parameters between agents) with one replay buffer. (b) Uncertainty estimates from an ensemble of neural networks on a toy regression task (see Appendix C for more experimental details). The black line is the ground truth curve, and the red dots are training samples. The blue lines show the mean and variance of predictions over ten ensemble models. The ensemble can produce well-calibrated uncertainty estimates (i.e., variance) on unseen samples.

**Soft Actor-Critic**. SAC (Haarnoja et al., 2018) is an off-policy actor-critic method based on the maximum entropy RL framework (Ziebart, 2010), which encourages the robustness to noise and exploration by maximizing a weighted objective of the reward and the policy entropy (see Appendix A for further details). To update the parameters, SAC alternates between a soft policy evaluation and a soft policy improvement. At the soft policy evaluation step, a soft Q-function, which is modeled as a neural network with parameters $\theta$, is updated by minimizing the following soft Bellman residual:

$$\mathcal{L}_{\texttt{critic}}^{\texttt{SAC}}(\theta) = \mathbb{E}_{\tau_t \sim \mathcal{B}}[\mathcal{L}_Q(\tau_t, \theta)], \tag{1}$$

$$\mathcal{L}_Q(\tau_t, \theta) = \left(Q_\theta(s_t, a_t) - r_t - \gamma \mathbb{E}_{a_{t+1} \sim \pi_\phi}\left[Q_{\bar{\theta}}(s_{t+1}, a_{t+1}) - \alpha \log \pi_\phi(a_{t+1}|s_{t+1})\right]\right)^2, \tag{2}$$

where $\tau_t = (s_t, a_t, r_t, s_{t+1})$ is a transition, $\mathcal{B}$ is a replay buffer, $\bar{\theta}$ are the delayed parameters, and $\alpha$ is a temperature parameter. At the soft policy improvement step, the policy $\pi$ with its parameter $\phi$ is updated by minimizing the following objective:

$$\mathcal{L}_{\texttt{actor}}^{\texttt{SAC}}(\phi) = \mathbb{E}_{s_t \sim \mathcal{B}}\left[\mathcal{L}_\pi(s_t, \phi)\right], \text{ where } \mathcal{L}_\pi(s_t, \phi) = \mathbb{E}_{a_t \sim \pi_\phi}\left[\alpha \log \pi_\phi(a_t|s_t) - Q_\theta(s_t, a_t)\right]. \tag{3}$$

Here, the policy is modeled as a Gaussian with mean and covariance given by neural networks to handle continuous action spaces.

## 4 SUNRISE

In this section, we propose the ensemble-based weighted Bellman backups, and then introduce SUN-RISE: **S**imple **UN**ified framework for **Re**Inforcement learning using en**SE**mbles, which combines various ensemble methods. In principle, our method can be used in conjunction with most modern off-policy RL algorithms, such as SAC (Haarnoja et al., 2018) and Rainbow DQN (Hessel et al., 2018). For the exposition, we describe only the SAC version in the main body. The Rainbow DQN version follows the same principles and is fully described in Appendix B.

### 4.1 WEIGHTED BELLMAN BACKUPS TO IMPROVE SIGNAL-TO-NOISE IN Q-UPDATES

Formally, we consider an ensemble of $N$ SAC agents, i.e., $\{Q_{\theta_i}, \pi_{\phi_i}\}_{i=1}^N$, where $\theta_i$ and $\phi_i$ denote the parameters of the $i$-th soft Q-function and policy.[1] Since conventional Q-learning is based on the Bellman backup in (2), it can be affected by error propagation. I.e., error in the target Q-function $Q_{\bar{\theta}}(s_{t+1}, a_{t+1})$ gets propagated into the Q-function $Q_\theta(s_t, a_t)$ at the current state. In other words, errors in the previous Q-function induce the "noise" to the learning "signal" (i.e., true Q-value) of the current Q-function. Recently, Kumar et al. (2020) showed that this error propagation can cause inconsistency and unstable convergence. To mitigate this issue, for each agent $i$, we consider a weighted Bellman backup as follows:

$$\mathcal{L}_{WQ}(\tau_t, \theta_i)$$
$$= w(s_{t+1}, a_{t+1})\left(Q_{\theta_i}(s_t, a_t) - r_t - \gamma\left(Q_{\bar{\theta}_i}(s_{t+1}, a_{t+1}) - \alpha \log \pi_\phi(a_{t+1}|s_{t+1})\right)\right)^2, \tag{4}$$

---

[1]We remark that each Q-function $Q_{\theta_i}(s, a)$ has a unique target Q-function $Q_{\bar{\theta}_i}(s, a)$.

where $\tau_t = (s_t, a_t, r_t, s_{t+1})$ is a transition, $a_{t+1} \sim \pi_\phi(a|s_t)$, and $w(s, a)$ is a confidence weight based on ensemble of target Q-functions:

$$w(s, a) = \sigma \left( -\bar{Q}_{\mathtt{std}}(s, a) * T \right) + 0.5, \tag{5}$$

where $T > 0$ is a temperature, $\sigma$ is the sigmoid function, and $\bar{Q}_{\mathtt{std}}(s, a)$ is the empirical standard deviation of all target Q-functions $\{Q_{\bar{\theta}_i}\}_{i=1}^N$. Note that the confidence weight is bounded in $[0.5, 1.0]$ because standard deviation is always positive.[2] The proposed objective $\mathcal{L}_{WQ}$ down-weights the sample transitions with high variance across target Q-functions, resulting in a loss function for the Q-updates that has a better signal-to-noise ratio.

### 4.2 COMBINATION WITH ADDITIONAL TECHNIQUES THAT LEVERAGE ENSEMBLES

We integrate the proposed weighted Bellman backup with UCB exploration into a single framework by utilizing the bootstrap with random initialization.

**Bootstrap with random initialization**. To train the ensemble of agents, we use the bootstrap with random initialization (Efron, 1982; Osband et al., 2016a), which enforces the diversity between agents through two simple ideas: First, we initialize the model parameters of all agents with random parameter values for inducing an initial diversity in the models. Second, we apply different samples to train each agent. Specifically, for each SAC agent $i$ in each timestep $t$, we draw the binary masks $m_{t,i}$ from the Bernoulli distribution with parameter $\beta \in (0, 1]$, and store them in the replay buffer. Then, when updating the model parameters of agents, we multiply the bootstrap mask to each objective function, such as: $m_{t,i}\mathcal{L}_\pi(s_t, \phi_i)$ and $m_{t,i}\mathcal{L}_{WQ}(\tau_t, \theta_i)$ in (3) and (4), respectively. We remark that Osband et al. (2016a) applied this simple technique to train an ensemble of DQN (Mnih et al., 2015) only for discrete control tasks, while we apply to SAC (Haarnoja et al., 2018) and Rainbow DQN (Hessel et al., 2018) for both continuous and discrete tasks with additional techniques.

**UCB exploration**. The ensemble can also be leveraged for efficient exploration (Chen et al., 2017; Osband et al., 2016a) because it can express higher uncertainty on unseen samples. Motivated by this, by following the idea of Chen et al. (2017), we consider an optimism-based exploration that chooses the action that maximizes

$$a_t = \max_a \{Q_{\mathtt{mean}}(s_t, a) + \lambda Q_{\mathtt{std}}(s_t, a)\}, \tag{6}$$

where $Q_{\mathtt{mean}}(s, a)$ and $Q_{\mathtt{std}}(s, a)$ are the empirical mean and standard deviation of all Q-functions $\{Q_{\theta_i}\}_{i=1}^N$, and the $\lambda > 0$ is a hyperparameter. This inference method can encourage exploration by adding an exploration bonus (i.e., standard deviation $Q_{\mathtt{std}}$) for visiting unseen state-action pairs similar to the UCB algorithm (Auer et al., 2002). We remark that this inference method was originally proposed in Chen et al. (2017) for efficient exploration in discrete action spaces. However, in continuous action spaces, finding the action that maximizes the UCB is not straightforward. To handle this issue, we propose a simple approximation scheme, which first generates $N$ candidate action set from ensemble policies $\{\pi_{\phi_i}\}_{i=1}^N$, and then chooses the action that maximizes the UCB (Line 4 in Algorithm 1). For evaluation, we approximate the maximum a posterior action by averaging the mean of Gaussian distributions modeled by each ensemble policy.

The full procedure of our unified framework, coined SUNRISE, is summarized in Algorithm 1.

## 5 EXPERIMENTAL RESULTS

We designed our experiments to answer the following questions:

- Can SUNRISE improve off-policy RL algorithms, such as SAC (Haarnoja et al., 2018) and Rainbow DQN (Hessel et al., 2018), for both continuous (see Table 1 and Table 2) and discrete (see Table 3) control tasks?
- How crucial is the proposed weighted Bellman backups in (4) for improving the signal-to-noise in Q-updates (see Figure 2)?
- Can UCB exploration be useful for solving tasks with sparse rewards (see Figure 3(b))?
- Is SUNRISE better than a single agent with more updates and parameters (see Figure 3(c))?
- How does ensemble size affect the performance (see Figure 3(d))?

---

[2]We find that it is empirically stable to set minimum value of weight $w(s, a)$ as 0.5.

---

**Algorithm 1** SUNRISE: SAC version

---

1: **for** each iteration **do**
2:     **for** each timestep $t$ **do**
3:         // UCB EXPLORATION
4:         Collect $N$ action samples: $\mathcal{A}_t = \{a_{t,i} \sim \pi_{\phi_i}(a|s_t)|i \in \{1, \ldots, N\}\}$
5:         Choose the action that maximizes UCB: $a_t = \underset{a_{t,i} \in \mathcal{A}_t}{\arg\max}\ Q_{\texttt{mean}}(s_t, a_{t,i}) + \lambda Q_{\texttt{std}}(s_t, a_{t,i})$
6:         Collect state $s_{t+1}$ and reward $r_t$ from the environment by taking action $a_t$
7:         Sample bootstrap masks $M_t = \{m_{t,i} \sim \text{Bernoulli}\,(\beta) - i \in \{1, \ldots, N\}\}$
8:         Store transitions $\tau_t = (s_t, a_t, s_{t+1}, r_t)$ and masks in replay buffer $\mathcal{B} \leftarrow \mathcal{B} \cup \{(\tau_t, M_t)\}$
9:     **end for**
10:     // UPDATE AGENTS VIA BOOTSTRAP AND WEIGHTED BELLMAN BACKUP
11:     **for** each gradient step **do**
12:         Sample random minibatch $\{(\tau_j, M_j)\}_{j=1}^{B} \sim \mathcal{B}$
13:         **for** each agent $i$ **do**
14:             Update the Q-function by minimizing $\frac{1}{B}\sum_{j=1}^{B} m_{j,i}\mathcal{L}_{WQ}\,(\tau_j, \theta_i)$ in (4)
15:             Update the policy by minimizing $\frac{1}{B}\sum_{j=1}^{B} m_{j,i}\mathcal{L}_{\pi}(s_j, \phi_i)$ in (3)
16:         **end for**
17:     **end for**
18: **end for**

---

## 5.1 SETUPS

**Continuous control tasks**. We evaluate SUNRISE on several continuous control tasks using simulated robots from OpenAI Gym (Brockman et al., 2016) and DeepMind Control Suite (Tassa et al., 2018). For OpenAI Gym experiments with proprioceptive inputs (e.g., positions and velocities), we compare to PETS (Chua et al., 2018), a state-of-the-art model-based RL method based on ensembles of dynamics models; POPLIN-P (Wang & Ba, 2020), a state-of-the-art model-based RL method which uses a policy network to generate actions for planning; POPLIN-A (Wang & Ba, 2020), variant of POPLIN-P which adds noise in the action space; METRPO (Kurutach et al., 2018), a hybrid RL method which augments TRPO (Schulman et al., 2015) using ensembles of dynamics models; and two state-of-the-art model-free RL methods, TD3 (Fujimoto et al., 2018) and SAC (Haarnoja et al., 2018). For our method, we consider a combination of SAC and SUNRISE, as described in Algorithm 1. Following the setup in Wang & Ba (2020) and Wang et al. (2019), we report the mean and standard deviation across ten runs after 200K timesteps on five complex environments: Cheetah, Walker, Hopper, Ant and SlimHumanoid with early termination (ET). More experimental details and learning curves with 1M timesteps are in Appendix D.

For DeepMind Control Suite with image inputs, we compare to PlaNet (Hafner et al., 2019), a model-based RL method which learns a latent dynamics model and uses it for planning; Dreamer (Hafner et al., 2020), a hybrid RL method which utilizes the latent dynamics model to generate synthetic roll-outs; SLAC (Lee et al., 2020), a hybrid RL method which combines the latent dynamics model with SAC; and three state-of-the-art model-free RL methods which apply contrastive learning (CURL; Srinivas et al. 2020) or data augmentation (RAD (Laskin et al., 2020) and DrQ (Kostrikov et al., 2020)) to SAC. For our method, we consider a combination of RAD (i.e., SAC with random crop) and SUNRISE. Following the setup in RAD, we report the mean and standard deviation across five runs after 100k (i.e., low sample regime) and 500k (i.e., asymptotically optimal regime) environment steps on six environments: Finger-spin, Cartpole-swing, Reacher-easy, Cheetah-run, Walker-walk, and Cup-catch. More experimental details and learning curves are in Appendix F.

**Discrete control benchmarks**. For discrete control tasks, we demonstrate the effectiveness of SUNRISE on several Atari games (Bellemare et al., 2013). We compare to SimPLe (Kaiser et al., 2020), a hybrid RL method which updates the policy only using samples generated by learned dynamics model; Rainbow DQN (Hessel et al., 2018) with modified hyperparameters for sample-efficiency (van Hasselt et al., 2019); Random agent (Kaiser et al., 2020); two state-of-the-art model-free RL methods which apply the contrastive learning (CURL; Srinivas et al. 2020) and data augmentation (DrQ; Kostrikov et al. 2020) to Rainbow DQN; and Human performances reported in Kaiser et al. (2020) and van Hasselt et al. (2019). Following the setups in SimPLe, we report the mean across

| | Cheetah | Walker | Hopper | Ant | SlimHumanoid-ET |
|---|---|---|---|---|---|
| PETS | $2288.4 \pm 1019.0$ | $282.5 \pm 501.6$ | $114.9 \pm 621.0$ | $1165.5 \pm 226.9$ | $\mathbf{2055.1 \pm 771.5}$ |
| POPLIN-A | $1562.8 \pm 1136.7$ | $-105.0 \pm 249.8$ | $202.5 \pm 962.5$ | $1148.4 \pm 438.3$ | - |
| POPLIN-P | $4235.0 \pm 1133.0$ | $597.0 \pm 478.8$ | $2055.2 \pm 613.8$ | $\mathbf{2330.1 \pm 320.9}$ | - |
| METRPO | $2283.7 \pm 900.4$ | $-1609.3 \pm 657.5$ | $1272.5 \pm 500.9$ | $282.2 \pm 18.0$ | $76.1 \pm 8.8$ |
| TD3 | $3015.7 \pm 969.8$ | $-516.4 \pm 812.2$ | $1816.6 \pm 994.8$ | $870.1 \pm 283.8$ | $1070.0 \pm 168.3$ |
| SAC | $4474.4 \pm 700.9$ | $299.5 \pm 921.9$ | $1781.3 \pm 737.2$ | $979.5 \pm 253.2$ | $1371.8 \pm 473.4$ |
| SUNRISE | $\mathbf{4501.8 \pm 443.8}$ | $\mathbf{1236.5 \pm 1123.9}$ | $\mathbf{2643.2 \pm 472.3}$ | $1502.4 \pm 483.5$ | $1926.6 \pm 375.0$ |

Table 1: Performance on OpenAI Gym at 200K timesteps. The results show the mean and standard deviation averaged over ten runs, and the best results are indicated in bold. For baseline methods, we report the best number in prior works (Wang & Ba, 2020; Wang et al., 2019).

| 500K step | PlaNet | Dreamer | SLAC | CURL | DrQ | RAD | SUNRISE |
|---|---|---|---|---|---|---|---|
| Finger-spin | $561 \pm 284$ | $796 \pm 183$ | $673 \pm 92$ | $926 \pm 45$ | $938 \pm 103$ | $975 \pm 16$ | $\mathbf{983} \pm 1$ |
| Cartpole-swing | $475 \pm 71$ | $762 \pm 27$ | - | $845 \pm 45$ | $868 \pm 10$ | $873 \pm 3$ | $\mathbf{876} \pm 4$ |
| Reacher-easy | $210 \pm 44$ | $793 \pm 164$ | - | $929 \pm 44$ | $942 \pm 71$ | $916 \pm 49$ | $\mathbf{982} \pm 3$ |
| Cheetah-run | $305 \pm 131$ | $570 \pm 253$ | $640 \pm 19$ | $518 \pm 28$ | $660 \pm 96$ | $624 \pm 10$ | $\mathbf{678} \pm 46$ |
| Walker-walk | $351 \pm 58$ | $897 \pm 49$ | $842 \pm 51$ | $902 \pm 43$ | $921 \pm 45$ | $938 \pm 9$ | $\mathbf{953} \pm 13$ |
| Cup-catch | $460 \pm 380$ | $879 \pm 87$ | $852 \pm 71$ | $959 \pm 27$ | $963 \pm 9$ | $966 \pm 9$ | $\mathbf{969} \pm 5$ |
| 100K step | | | | | | | |
| Finger-spin | $136 \pm 216$ | $341 \pm 70$ | $693 \pm 141$ | $767 \pm 56$ | $901 \pm 104$ | $811 \pm 146$ | $\mathbf{905} \pm 57$ |
| Cartpole-swing | $297 \pm 39$ | $326 \pm 27$ | - | $582 \pm 146$ | $\mathbf{759} \pm 92$ | $373 \pm 90$ | $591 \pm 55$ |
| Reacher-easy | $20 \pm 50$ | $314 \pm 155$ | - | $538 \pm 233$ | $601 \pm 213$ | $567 \pm 54$ | $\mathbf{722} \pm 50$ |
| Cheetah-run | $138 \pm 88$ | $235 \pm 137$ | $319 \pm 56$ | $299 \pm 48$ | $344 \pm 67$ | $381 \pm 79$ | $\mathbf{413} \pm 35$ |
| Walker-walk | $224 \pm 48$ | $277 \pm 12$ | $361 \pm 73$ | $403 \pm 24$ | $612 \pm 164$ | $641 \pm 89$ | $\mathbf{667} \pm 147$ |
| Cup-catch | $0 \pm 0$ | $246 \pm 174$ | $512 \pm 110$ | $769 \pm 43$ | $\mathbf{913} \pm 53$ | $666 \pm 181$ | $633 \pm 241$ |

Table 2: Performance on DeepMind Control Suite at 100K and 500K environment steps. The results show the mean and standard deviation averaged five runs, and the best results are indicated in bold. For baseline methods, we report the best numbers reported in prior works (Kostrikov et al., 2020).

three runs after 100K interactions (i.e., 400K frames with action repeat of 4). For our method, we consider a combination of sample-efficient versions of Rainbow DQN and SUNRISE (see Algorithm 3 in Appendix B). More experimental details and learning curves are in Appendix G.

For our method, we do not alter any hyperparameters of the original RL algorithms and train five ensemble agents. There are only three additional hyperparameters $\beta$, $T$, and $\lambda$ for bootstrap, weighted Bellman backup, and UCB exploration, where we provide details in Appendix D, F, and G.

## 5.2 COMPARATIVE EVALUATION

**OpenAI Gym**. Table 1 shows the average returns of evaluation roll-outs for all methods. SUNRISE consistently improves the performance of SAC across all environments and outperforms the model-based RL methods, such as POPLIN-P and PETS, on all environments except Ant and SlimHumanoid-ET. Even though we focus on performance after small samples because of the recent emphasis on making RL more sample efficient, we find that the gain from SUNRISE becomes even more significant when training longer (see Figure 3(c) and Appendix D). We remark that SUNRISE is more compute-efficient than modern model-based RL methods, such as POPLIN and PETS, because they also utilize ensembles (of dynamics models) and perform planning to select actions. Namely, SUNRISE is simple to implement, computationally efficient, and readily parallelizable.

**DeepMind Control Suite**. As shown in Table 2, SUNRISE also consistently improves the performance of RAD (i.e., SAC with random crop) on all environments from DeepMind Control Suite. This implies that the proposed method can be useful for high-dimensional and complex input observations. Moreover, our method outperforms existing pixel-based RL methods in almost all environments. We remark that SUNRISE can also be combined with DrQ, and expect that it can achieve better performances on Cartpole-swing and Cup-catch at 100K environment steps.

| Game | Human | Random | SimPLe | CURL | DrQ | Rainbow | SUNRISE |
|---|---|---|---|---|---|---|---|
| Alien | 7127.7 | 227.8 | 616.9 | 558.2 | 761.4 | 789.0 | **872.0** |
| Amidar | 1719.5 | 5.8 | 88.0 | **142.1** | 97.3 | 118.5 | 122.6 |
| Assault | 742.0 | 222.4 | 527.2 | **600.6** | 489.1 | 413.0 | 594.8 |
| Asterix | 8503.3 | 210.0 | **1128.3** | 734.5 | 637.5 | 533.3 | 755.0 |
| BankHeist | 753.1 | 14.2 | 34.2 | 131.6 | 196.6 | 97.7 | **266.7** |
| BattleZone | 37187.5 | 2360.0 | 5184.4 | 14870.0 | 13520.6 | 7833.3 | **15700.0** |
| Boxing | 12.1 | 0.1 | **9.1** | 1.2 | 6.9 | 0.6 | 6.7 |
| Breakout | 30.5 | 1.7 | **16.4** | 4.9 | 14.5 | 2.3 | 1.8 |
| ChopperCommand | 7387.8 | 811.0 | **1246.9** | 1058.5 | 646.6 | 590.0 | 1040.0 |
| CrazyClimber | 35829.4 | 10780.5 | **62583.6** | 12146.5 | 19694.1 | 25426.7 | 22230.0 |
| DemonAttack | 1971.0 | 152.1 | 208.1 | 817.6 | **1222.2** | 688.2 | 919.8 |
| Freeway | 29.6 | 0.0 | 20.3 | 26.7 | 15.4 | 28.7 | **30.2** |
| Frostbite | 4334.7 | 65.2 | 254.7 | 1181.3 | 449.7 | 1478.3 | **2026.7** |
| Gopher | 2412.5 | 257.6 | 771.0 | **669.3** | 598.4 | 348.7 | 654.7 |
| Hero | 30826.4 | 1027.0 | 2656.6 | 6279.3 | 4001.6 | 3675.7 | **8072.5** |
| Jamesbond | 302.8 | 29.0 | 125.3 | **471.0** | 272.3 | 300.0 | 390.0 |
| Kangaroo | 3035.0 | 52.0 | 323.1 | 872.5 | 1052.4 | 1060.0 | **2000.0** |
| Krull | 2665.5 | 1598.0 | **4539.9** | 4229.6 | 4002.3 | 2592.1 | 3087.2 |
| KungFuMaster | 22736.3 | 258.5 | **17257.2** | 14307.8 | 7106.4 | 8600.0 | 10306.7 |
| MsPacman | 6951.6 | 307.3 | 1480.0 | 1465.5 | 1065.6 | 1118.7 | **1482.3** |
| Pong | 14.6 | -20.7 | **12.8** | -16.5 | -11.4 | -19.0 | -19.3 |
| PrivateEye | 69571.3 | 24.9 | 58.3 | **218.4** | 49.2 | 97.8 | 100.0 |
| Qbert | 13455.0 | 163.9 | 1288.8 | 1042.4 | 1100.9 | 646.7 | **1830.8** |
| RoadRunner | 7845.0 | 11.5 | 5640.6 | 5661.0 | 8069.8 | 9923.3 | **11913.3** |
| Seaquest | 42054.7 | 68.4 | **683.3** | 384.5 | 321.8 | 396.0 | 570.7 |
| UpNDown | 11693.2 | 533.4 | 3350.3 | 2955.2 | 3924.9 | 3816.0 | **5074.0** |

Table 3: Performance on Atari games at 100K interactions. The results show the scores averaged three runs, and the best results are indicated in bold. For baseline methods, we report the best numbers reported in prior works (Kaiser et al., 2020; van Hasselt et al., 2019).

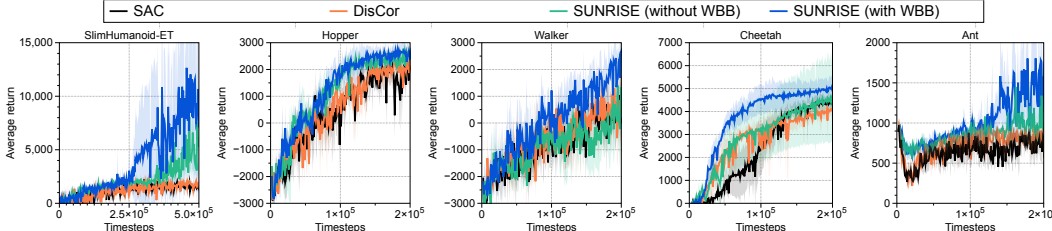

Figure 2: Learning curves on OpenAI Gym with noisy rewards. To verify the effects of the weighted Bellman backups (WBB), we consider SUNRISE with WBB and without WBB. The solid line and shaded regions represent the mean and standard deviation, respectively, across four runs.

**Atari games**. We also evaluate SUNRISE on discrete control tasks from the Atari benchmark using Rainbow DQN. Table 3 shows that SUNRISE improves the performance of Rainbow in almost all environments, and outperforms the state-of-the-art CURL and SimPLe on 11 out of 26 Atari games. Here, we remark that SUNRISE is also compatible with CURL, which could enable even better performance. These results demonstrate that SUNRISE is a general approach.

## 5.3 ABLATION STUDY

**Effects of weighted Bellman backups**. To verify the effectiveness of the proposed weighted Bell-man backup (4) in improving signal-to-noise in Q-updates, we evaluate on a modified OpenAI Gym environments with noisy rewards. Following Kumar et al. (2019), we add Gaussian noise to the reward function: $r'(s, a) = r(s, a) + z$, where $z \sim \mathcal{N}(0, 1)$ only during training, and report the deterministic ground-truth reward during evaluation. For our method, we also consider a variant of SUNRISE, which updates Q-functions without the proposed weighted Bellman backup to isolate its effect. We compare to DisCor (Kumar et al., 2020), which improves SAC by reweighting the Bellman backup based on estimated cumulative Bellman errors (see Appendix E for more details).

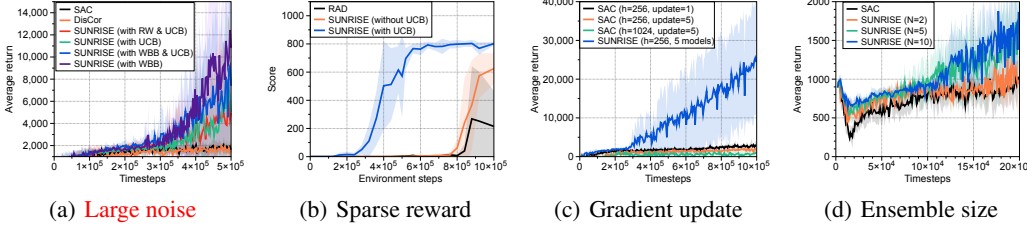

| (a) Large noise | (b) Sparse reward | (c) Gradient update | (d) Ensemble size |

Figure 3: (a) Learning curves of SUNRISE with random weight (RW) and the proposed weighted Bellman backups (WBB) on the SlimHumanoid-ET environment with noisy rewards. (b) Effects of UCB exploration on the Cartpole environment with sparse reward. (c) Learning curves of SUN-RISE and single agent with $h$ hidden units and five gradient updates per each timestep on the SlimHumanoid-ET environment. (d) Learning curves of SUNRISE with varying values of ensemble size $N$ on the Ant environment.

Figure 2 shows the learning curves of all methods on OpenAI Gym with noisy rewards. The proposed weighted Bellman backup significantly improves both sample-efficiency and asymptotic performance of SUNRISE, and outperforms baselines such as SAC and DisCor. One can note the performance gain due to our weighted Bellman backup becomes more significant in complex environments, such as SlimHumanoid-ET. We remark that DisCor still suffers from error propagation issues in complex environments like SlimHumanoid-ET and Ant because there are some approximation errors in estimating cumulative Bellman errors (see Section 6.1 for more detailed discussion). These results imply that errors in the target Q-function can be characterized by the proposed confident weight in equation 5 effectively.

We also consider another variant of SUNRISE, which updates Q-functions with random weights sampled from $[0.5, 1.0]$ uniformly at random. In order to evaluate the performance of SUNRISE, we increase the noise rate by adding Gaussian noise with a large standard deviation to the reward function: $r'(s, a) = r(s, a) + z$, where $z \sim \mathcal{N}(0, 5)$. Figure 3(a) shows the learning curves of all methods on the SlimHumanoid-ET environment over 10 random seeds. First, one can not that SUNRISE with random weights (red curve) is worse than SUNRISE with the proposed weighted Bellman backups (blue curve). Additionally, even without UCB exploration, SUNRISE with the proposed weighted Bellman backups (purple curve) outperforms all baselines. This implies that the proposed weighted Bellman backups can handle the error propagation effectively even though there is a large noise in reward function.

**Effects of UCB exploration**. To verify the advantage of UCB exploration in (6), we evaluate on Cartpole-swing with sparse-reward from DeepMind Control Suite. For our method, we consider a variant of SUNRISE, which selects action without UCB exploration. As shown in Fig 3(b), SUNRISE with UCB exploration (blue curve) significantly improves the sample-efficiency on the environment with sparse rewards.

**Comparison with a single agent with more updates/parameters**. One concern in utilizing the ensemble method is that its gains may come from more gradient updates and parameters. To clarify this concern, we compare SUNRISE (5 ensembles using 2-layer MLPs with 256 hidden units each) to a single agent, which consists of 2-layer MLPs with 1024 (and 256) hidden units with 5 updates using different random minibatches. Figure 3(c) shows that the learning curves on SlimHumanoid-ET, where SUNRISE outperforms all baselines. This implies that the gains from SUNRISE can not be achieved by simply increasing the number of updates/parameters. More experimental results on other environments are also available in Appendix D.

**Effects of ensemble size**. We analyze the effects of ensemble size $N$ on the Ant environment from OpenAI Gym. Figure 3(d) shows that the performance can be improved by increasing the ensemble size, but the improvement is saturated around $N = 5$. Thus, we use five ensemble agents for all experiments. More experimental results on other environments are also available in Appendix D, where the overall trend is similar.

## 6 DISCUSSION

### 6.1 CONNECTION WITH DISCOR

Kumar et al. (2020) show that naive Bellman backups can suffer from slow learning in certain environments, requiring exponentially many updates. To handle this problem, they propose the weighted Bellman backups, which make steady learning progress by inducing some optimal data distribution (see (Kumar et al., 2020) for more details). Specifically, in addition to a standard Q-learning, DisCor trains an error model $\Delta_\psi(s, a)$, which approximates the cumulative sum of discounted Bellman errors over the past iterations of training. Then, using the error model, DisCor reweights the Bellman backups based on a confidence weight defined as follows: $w(s, a) \propto \exp\left(-\frac{\gamma\Delta_\psi(s,a)}{T}\right)$, where $\gamma$ is a discount factor and $T$ is a temperature.

However, we remark that DisCor can still suffer from the error propagation issues because there is also an approximation error in estimating cumulative Bellman errors. Therefore, we consider an alternative approach that utilizes the uncertainty from ensembles. Because it has been observed that the ensemble can produce well-calibrated uncertainty estimates (i.e., variance) on unseen samples (Lakshminarayanan et al., 2017), we expect that the weighted Bellman backups based on ensembles can handle error propagation more effectively. Indeed, in our experiments, we find that ensemble-based weighted Bellman backups can give rise to more stable training and improve the data-efficiency of various off-policy RL algorithms.

### 6.2 COMPUTATION OVERHEAD

One can expect that there is an additional computation overhead by introducing ensembles. When we have $N$ ensemble agents, our method requires $N\times$ inferences for weighted Bellman backups and $2N\times$ inferences ($N$ for actors and $N$ for critics). However, we remark that our method can be more computationally efficient because it is parallelizable. Also, as shown in Figure 3(c), the gains from SUNRISE can not be achieved by simply increasing the number of updates/parameters.

## 7 CONCLUSION

In this paper, we present the ensemble-based weighted Bellman backups, which is compatible with various off-policy RL algorithms. By re-weighting target Q-values based on uncertainty estimates, we stabilize and improve the learning process on both continuous and discrete control benchmarks. Additionally, we introduce SUNRISE, a simple unified ensemble method, which integrates the proposed weighted Bellman backups with bootstrap with random initialization, and UCB exploration to handle various issues in off-policy RL algorithms. Our experiments show that SUNRISE consistently improves the performances of existing off-policy RL algorithms, such as Soft Actor-Critic and Rainbow DQN, and outperforms state-of-the-art RL algorithms for both continuous and discrete control tasks on both low-dimensional and high-dimensional environments. We hope that SUNRISE could be useful to other relevant topics such as sim-to-real transfer (Tobin et al., 2017), imitation learning (Torabi et al., 2018), understanding the connection between on-policy and off-policy RL (Schulman et al., 2017), offline RL (Agarwal et al., 2020), and planning (Srinivas et al., 2018; Tamar et al., 2016).

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

# Appendix

## A SUNRISE: SOFT ACTOR-CRITIC

**Background**. SAC (Haarnoja et al., 2018) is a state-of-the-art off-policy algorithm for continuous control problems. SAC learns a policy, $\pi_\phi(a|s)$, and a critic, $Q_\theta(s,a)$, and aims to maximize a weighted objective of the reward and the policy entropy, $\mathbb{E}_{s_t, a_t \sim \pi}\left[\sum_t \gamma^{t-1} r_t + \alpha \mathcal{H}(\pi_\phi(\cdot|s_t))\right]$. To update the parameters, SAC alternates between a soft policy evaluation and a soft policy improvement. At the soft policy evaluation step, a soft Q-function, which is modeled as a neural network with parameters $\theta$, is updated by minimizing the following soft Bellman residual:

$$\mathcal{L}_{\text{critic}}^{\text{SAC}}(\theta) = \mathbb{E}_{\tau_t \sim \mathcal{B}}[\mathcal{L}_Q(\tau_t, \theta)],$$

$$\mathcal{L}_Q(\tau_t, \theta) = \left(Q_\theta(s_t, a_t) - r_t - \gamma \mathbb{E}_{a_{t+1} \sim \pi_\phi}\left[Q_{\bar{\theta}}(s_{t+1}, a_{t+1}) - \alpha \log \pi_\phi(a_{t+1}|s_{t+1})\right]\right)^2,$$

where $\tau_t = (s_t, a_t, r_t, s_{t+1})$ is a transition, $\mathcal{B}$ is a replay buffer, $\bar{\theta}$ are the delayed parameters, and $\alpha$ is a temperature parameter. At the soft policy improvement step, the policy $\pi$ with its parameter $\phi$ is updated by minimizing the following objective:

$$\mathcal{L}_{\text{actor}}^{\text{SAC}}(\phi) = \mathbb{E}_{s_t \sim \mathcal{B}}\left[\mathcal{L}_\pi(s_t, \phi)\right], \text{ where } \mathcal{L}_\pi(s_t, \phi) = \mathbb{E}_{a_t \sim \pi_\phi}\left[\alpha \log \pi_\phi(a_t|s_t) - Q_\theta(s_t, a_t)\right].$$

We remark that this corresponds to minimizing the Kullback-Leibler divergence between the policy and a Boltzmann distribution induced by the current soft Q-function.

**SUNRISE without UCB exploration**. For SUNRISE without UCB exploration, we use random inference proposed in Bootstrapped DQN (Osband et al., 2016a), which randomly selects an index of policy uniformly at random and generates the action from the selected actor for the duration of that episode (see Line 3 in Algorithm 2).

---

**Algorithm 2** SUNRISE: SAC version (random inference)

---

1: **for** each iteration **do**
2:     // RANDOM INFERENCE
3:     Select an index of policy using $\widehat{i} \sim \text{Uniform}\{1, \cdots, N\}$
4:     **for** each timestep $t$ **do**
5:         Get the action from selected policy: $a_t \sim \pi_{\phi_{\widehat{i}}}(a|s_t)$
6:         Collect state $s_{t+1}$ and reward $r_t$ from the environment by taking action $a_t$
7:         Sample bootstrap masks $M_t = \{m_{t,i} \sim \text{Bernoulli}\,(\beta) - i \in \{1, \ldots, N\}\}$
8:         Store transitions $\tau_t = (s_t, a_t, s_{t+1}, r_t)$ and masks in replay buffer $\mathcal{B} \leftarrow \mathcal{B} \cup \{(\tau_t, M_t)\}$
9:     **end for**
10:     // UPDATE AGENTS VIA BOOTSTRAP AND WEIGHTED BELLMAN BACKUP
11:     **for** each gradient step **do**
12:         Sample random minibatch $\{(\tau_j, M_j)\}_{j=1}^B \sim \mathcal{B}$
13:         **for** each agent $i$ **do**
14:             Update the Q-function by minimizing $\frac{1}{B} \sum_{j=1}^B m_{j,i} \mathcal{L}_{WQ}(\tau_j, \theta_i)$
15:             Update the policy by minimizing $\frac{1}{B} \sum_{j=1}^B m_{j,i} \mathcal{L}_\pi(s_j, \phi_i)$
16:         **end for**
17:     **end for**
18: **end for**

---

## B EXTENSION TO RAINBOW DQN

### B.1 PRELIMINARIES: RAINBOW DQN

**Background**. DQN algorithm (Mnih et al., 2015) learns a Q-function, which is modeled as a neural network with parameters $\theta$, by minimizing the following Bellman residual:

$$\mathcal{L}^{\text{DQN}}(\theta) = \mathbb{E}_{\tau_t \sim \mathcal{B}}\left[\left(Q_\theta(s_t, a_t) - r_t - \gamma \max_a Q_{\bar{\theta}}(s_{t+1}, a)\right)^2\right], \tag{7}$$

where $\tau_t = (s_t, a_t, r_t, s_{t+1})$ is a transition, $\mathcal{B}$ is a replay buffer, and $\bar{\theta}$ are the delayed parameters. Even though Rainbow DQN integrates several techniques, such as double Q-learning (Van Hasselt et al., 2016) and distributional DQN (Bellemare et al., 2017), applying SUNRISE to Rainbow DQN can be described based on the standard DQN algorithm. For exposition, we refer the reader to Hessel et al. (2018) for more detailed explanations of Rainbow DQN.

---

**Algorithm 3** SUNRISE: Rainbow version

---

 1: **for** each iteration **do**
 2:     **for** each timestep $t$ **do**
 3:         // UCB EXPLORATION
 4:         Choose the action that maximizes UCB: $a_t = \arg\max_{a_{t,i} \in \mathcal{A}} Q_{\mathtt{mean}}(s_t, a_{t,i}) + \lambda Q_{\mathtt{std}}(s_t, a_{t,i})$
 5:         Collect state $s_{t+1}$ and reward $r_t$ from the environment by taking action $a_t$
 6:         Sample bootstrap masks $M_t = \{m_{t,i} \sim \text{Bernoulli}\,(\beta) - i \in \{1, \ldots, N\}\}$
 7:         Store transitions $\tau_t = (s_t, a_t, s_{t+1}, r_t)$ and masks in replay buffer $\mathcal{B} \leftarrow \mathcal{B} \cup \{(\tau_t, M_t)\}$
 8:     **end for**
 9:     // UPDATE Q-FUNCTIONS VIA BOOTSTRAP AND WEIGHTED BELLMAN BACKUP
10:     **for** each gradient step **do**
11:         Sample random minibatch $\{(\tau_j, M_j)\}_{j=1}^B \sim \mathcal{B}$
12:         **for** each agent $i$ **do**
13:             Update the Q-function by minimizing $\frac{1}{B} \sum_{j=1}^B m_{j,i} \mathcal{L}_{WQ}^{\mathtt{DQN}}(\tau_j, \theta_i)$
14:         **end for**
15:     **end for**
16: **end for**

---

### B.2 SUNRISE: RAINBOW DQN

**Bootstrap with random initialization**. Formally, we consider an ensemble of $N$ Q-functions, i.e., $\{Q_{\theta_i}\}_{i=1}^N$, where $\theta_i$ denotes the parameters of the $i$-th Q-function.[3] To train the ensemble of Q-functions, we use the bootstrap with random initialization (Efron, 1982; Osband et al., 2016a), which enforces the diversity between Q-functions through two simple ideas: First, we initialize the model parameters of all Q-functions with random parameter values for inducing an initial diversity in the models. Second, we apply different samples to train each Q-function. Specifically, for each Q-function $i$ in each timestep $t$, we draw the binary masks $m_{t,i}$ from the Bernoulli distribution with parameter $\beta \in (0, 1]$, and store them in the replay buffer. Then, when updating the model parameters of Q-functions, we multiply the bootstrap mask to each objective function.

**Weighted Bellman backup**. Since conventional Q-learning is based on the Bellman backup in equation 7, it can be affected by error propagation. I.e., error in the target Q-function $Q_{\bar{\theta}}(s_{t+1}, a_{t+1})$ gets propagated into the Q-function $Q_\theta(s_t, a_t)$ at the current state. Recently, Kumar et al. (2020) showed that this error propagation can cause inconsistency and unstable convergence. To mitigate this issue, for each Q-function $i$, we consider a weighted Bellman backup as follows:

$$\mathcal{L}_{WQ}^{\mathtt{DQN}}(\tau_t, \theta_i) = w(s_{t+1}) \left( Q_{\theta_i}(s_t, a_t) - r_t - \gamma \max_a Q_{\bar{\theta}_i}(s_{t+1}, a) \right)^2,$$

where $\tau_t = (s_t, a_t, r_t, s_{t+1})$ is a transition, and $w(s)$ is a confidence weight based on ensemble of target Q-functions:

$$w(s) = \sigma\left( -\bar{Q}_{\mathtt{std}}(s) * T \right) + 0.5, \tag{8}$$

where $T > 0$ is a temperature, $\sigma$ is the sigmoid function, and $\bar{Q}_{\mathtt{std}}(s)$ is the empirical standard deviation of all target Q-functions $\{\max_a Q_{\bar{\theta}_i}(s, a)\}_{i=1}^N$. Note that the confidence weight is bounded in $[0.5, 1.0]$ because standard deviation is always positive.[4] The proposed objective $\mathcal{L}_{WQ}^{\mathtt{DQN}}$ down-weights the sample transitions with high variance across target Q-functions, resulting in a loss function for the Q-updates that has a better signal-to-noise ratio. Note that we combine the proposed

---

[3]Here, we remark that each Q-function has a unique target Q-function.
[4]We find that it is empirically stable to set minimum value of weight $w(s, a)$ as 0.5.

weighted Bellman backup with prioritized replay (Schaul et al., 2016) by multiplying both weights to Bellman backups.

**UCB exploration**. The ensemble can also be leveraged for efficient exploration (Chen et al., 2017; Osband et al., 2016a) because it can express higher uncertainty on unseen samples. Motivated by this, by following the idea of Chen et al. (2017), we consider an optimism-based exploration that chooses the action that maximizes

$$a_t = \max_a \{Q_{\mathtt{mean}}(s_t, a) + \lambda Q_{\mathtt{std}}(s_t, a)\}, \tag{9}$$

where $Q_{\mathtt{mean}}(s, a)$ and $Q_{\mathtt{std}}(s, a)$ are the empirical mean and standard deviation of all Q-functions $\{Q_{\theta_i}\}_{i=1}^N$, and the $\lambda > 0$ is a hyperparameter. This inference method can encourage exploration by adding an exploration bonus (i.e., standard deviation $Q_{\mathtt{std}}$) for visiting unseen state-action pairs similar to the UCB algorithm (Auer et al., 2002). This inference method was originally proposed in Chen et al. (2017) for efficient exploration in DQN, but we further extend it to Rainbow DQN. For evaluation, we approximate the maximum a posterior action by choosing the action maximizes the mean of Q-functions, i.e., $a_t = \max_a \{Q_{\mathtt{mean}}(s_t, a)\}$. The full procedure is summarized in Algorithm 3.

## C   IMPLEMENTATION DETAILS FOR TOY REGRESSION TASKS

We evaluate the quality of uncertainty estimates from an ensemble of neural networks on a toy regression task. To this end, we generate twenty training samples drawn as $y = x^3 + \epsilon$, where $\epsilon \sim \mathcal{N}(0, 3^2)$, and train ten ensembles of regression networks using bootstrap with random initialization. The regression network is as fully-connected neural networks with 2 hidden layers and 50 rectified linear units in each layer. For bootstrap, we draw the binary masks from the Bernoulli distribution with mean $\beta = 0.3$. As uncertainty estimates, we measure the empirical variance of the networks' predictions. As shown in Figure 1(b), the ensemble can produce well-calibrated uncertainty estimates (i.e., variance) on unseen samples.

## D   EXPERIMENTAL SETUPS AND RESULTS: OPENAI GYM

**Environments**. We evaluate the performance of SUNRISE on four complex environments based on the standard bench-marking environments[5] from OpenAI Gym (Brockman et al., 2016). Note that we do not use a modified Cheetah environments from PETS (Chua et al., 2018) (dented as Cheetah in POPLIN (Wang & Ba, 2020)) because it includes additional information in observations.

**Training details**. We consider a combination of SAC and SUNRISE using the publicly released implementation repository (https://github.com/vitchyr/rlkit) without any modifications on hyperparameters and architectures. For our method, the temperature for weighted Bellman backups is chosen from $T \in \{10, 20, 50\}$, the mean of the Bernoulli distribution is chosen from $\beta \in \{0.5, 1.0\}$, the penalty parameter is chosen from $\lambda \in \{1, 5, 10\}$, and we train five ensemble agents. The optimal parameters are chosen to achieve the best performance on training environments. Here, we remark that training ensemble agents using same training samples but with different initialization (i.e., $\beta = 1$) usually achieves the best performance in most cases similar to Osband et al. (2016a) and Chen et al. (2017). We expect that this is because splitting samples can reduce the sample-efficiency. Also, initial diversity from random initialization can be enough because each Q-function has a unique target Q-function, i.e., target value is also different according to initialization.

**Learning curves**. Figure 4 shows the learning curves on all environments. One can note that SUNRISE consistently improves the performance of SAC by a large margin.

**Effects of ensembles**. Figure 5 shows the learning curves of SUNRISE with varying values of ensemble size on all environments. The performance can be improved by increasing the ensemble size, but the improvement is saturated around $N = 5$.

---

[5]We used the reference implementation at https://github.com/WilsonWangTHU/mbbl (Wang et al., 2019).

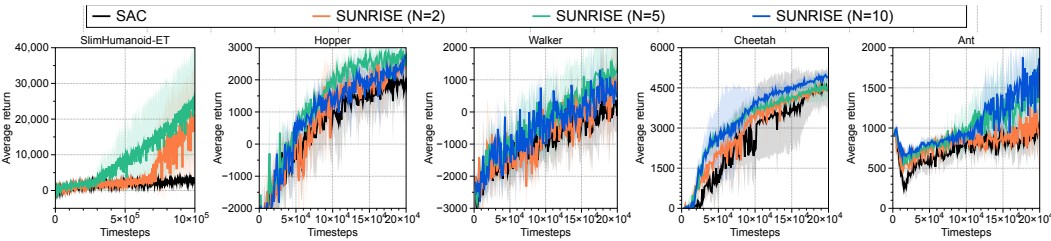

Figure 4: Learning curves of SUNRISE and single agent with $h$ hidden units and five gradient updates per each timestep on OpenAI Gym. The solid line and shaded regions represent the mean and standard deviation, respectively, across four runs.

Figure 5: Learning curves of SUNRISE with varying values of ensemble size $N$. The solid line and shaded regions represent the mean and standard deviation, respectively, across four runs.

## E  EXPERIMENTAL SETUPS AND RESULTS: NOISY REWARD

**DisCor**. DisCor (Kumar et al., 2020) was proposed to prevent the error propagation issue in Q-learning. In addition to a standard Q-learning, DisCor trains an error model $\Delta_\psi(s, a)$, which approximates the cumulative sum of discounted Bellman errors over the past iterations of training. Then, using the error model, DisCor reweights the Bellman backups based on a confidence weight defined as follows:

$$w(s, a) \propto \exp\left(-\frac{\gamma\Delta_\psi(s, a)}{T}\right),$$

where $\gamma$ is a discount factor and $T$ is a temperature. By following the setups in Kumar et al. (2020), we take a network with 1 extra hidden layer than the corresponding Q-network as an error model, and chose $T = 10$ for all experiments. We update the temperature via a moving average and use the learning rate of $0.0003$. We use the SAC algorithm as the RL objective coupled with DisCor and build on top of the publicly released implementation repository (https://github.com/vitchyr/rlkit).

| Hyperparameter | Value | Hyperparameter | Value |
|---|---|---|---|
| Random crop | True | Initial temperature | 0.1 |
| Observation rendering | $(100, 100)$ | Learning rate $(f_\theta, \pi_\psi, Q_\phi)$ | $2e-4$ cheetah, run |
| Observation downsampling | $(84, 84)$ | | $1e-3$ otherwise |
| Replay buffer size | 100000 | Learning rate $(\alpha)$ | $1e-4$ |
| Initial steps | 1000 | Batch Size | 512 (cheetah), 256 (rest) |
| Stacked frames | 3 | $Q$ function EMA $\tau$ | 0.01 |
| Action repeat | 2 finger, spin; walker, walk | Critic target update freq | 2 |
| | 8 cartpole, swingup | Convolutional layers | 4 |
| | 4 otherwise | Number of filters | 32 |
| Hidden units (MLP) | 1024 | Non-linearity | ReLU |
| Evaluation episodes | 10 | Encoder EMA $\tau$ | 0.05 |
| Optimizer | Adam | Latent dimension | 50 |
| $(\beta_1, \beta_2) \to (f_\theta, \pi_\psi, Q_\phi)$ | $(.9, .999)$ | Discount $\gamma$ | .99 |
| $(\beta_1, \beta_2) \to (\alpha)$ | $(.5, .999)$ | | |

Table 4: Hyperparameters used for DeepMind Control Suite experiments. Most hyperparameters values are unchanged across environments with the exception for action repeat, learning rate, and batch size.

## F  EXPERIMENTAL SETUPS AND RESULTS: DEEPMIND CONTROL SUITE

**Training details**. We consider a combination of RAD and SUNRISE using the publicly released implementation repository (`https://github.com/MishaLaskin/rad`) with a full list of hyperparameters in Table 4. Similar to Laskin et al. (2020), we use the same encoder architecture as in (Yarats et al., 2019), and the actor and critic share the same encoder to embed image observations.[6] For our method, the temperature for weighted Bellman backups is chosen from $T \in \{10, 100\}$, the mean of the Bernoulli distribution is chosen from $\beta \in \{0.5, 1.0\}$, the penalty parameter is chosen from $\lambda \in \{1, 5, 10\}$, and we train five ensemble agents. The optimal parameters are chosen to achieve the best performance on training environments. Here, we remark that training ensemble agents using same training samples but with different initialization (i.e., $\beta = 1$) usually achieves the best performance in most cases similar to Osband et al. (2016a) and Chen et al. (2017). We expect that this is because training samples can reduce the sample-efficiency. Also, initial diversity from random initialization can be enough because each Q-function has a unique target Q-function, i.e., target value is also different according to initialization.

**Learning curves**. Figure 6(g), 6(h), 6(i), 6(j), 6(k), and 6(l) show the learning curves on all environments. Since RAD already achieves the near optimal performances and the room for improvement is small, we can see a small but consistent gains from SUNRISE. To verify the effectiveness of SUN-RISE more clearly, we consider a combination of SAC and SUNRISE in Figure 6(a), 6(b), 6(c), 6(d), 6(e), and 6(f), where the gain from SUNRISE is more significant.

## G  EXPERIMENTAL SETUPS AND RESULTS: ATARI GAMES

**Training details**. We consider a combination of sample-efficient versions of Rainbow DQN and SUNRISE using the publicly released implementation repository (`https://github.com/Kaixhin/Rainbow`) without any modifications on hyperparameters and architectures. For our method, the temperature for weighted Bellman backups is chosen from $T \in \{10, 40\}$, the mean of the Bernoulli distribution is chosen from $\beta \in \{0.5, 1.0\}$, the penalty parameter is chosen from $\lambda \in \{1, 10\}$, and we train five ensemble agents. The optimal parameters are chosen to achieve the best performance on training environments. Here, we remark that training ensemble agents using same training samples but with different initialization (i.e., $\beta = 1$) usually achieves the best performance in most cases similar to Osband et al. (2016a) and Chen et al. (2017). We expect that this is because splitting samples can reduce the sample-efficiency. Also, initial diversity from random initialization can be enough because each Q-function has a unique target Q-function, i.e., target value is also different according to initialization.

---

[6]However, we remark that each agent does not share the encoders unlike Bootstrapped DQN (Osband et al., 2016a).

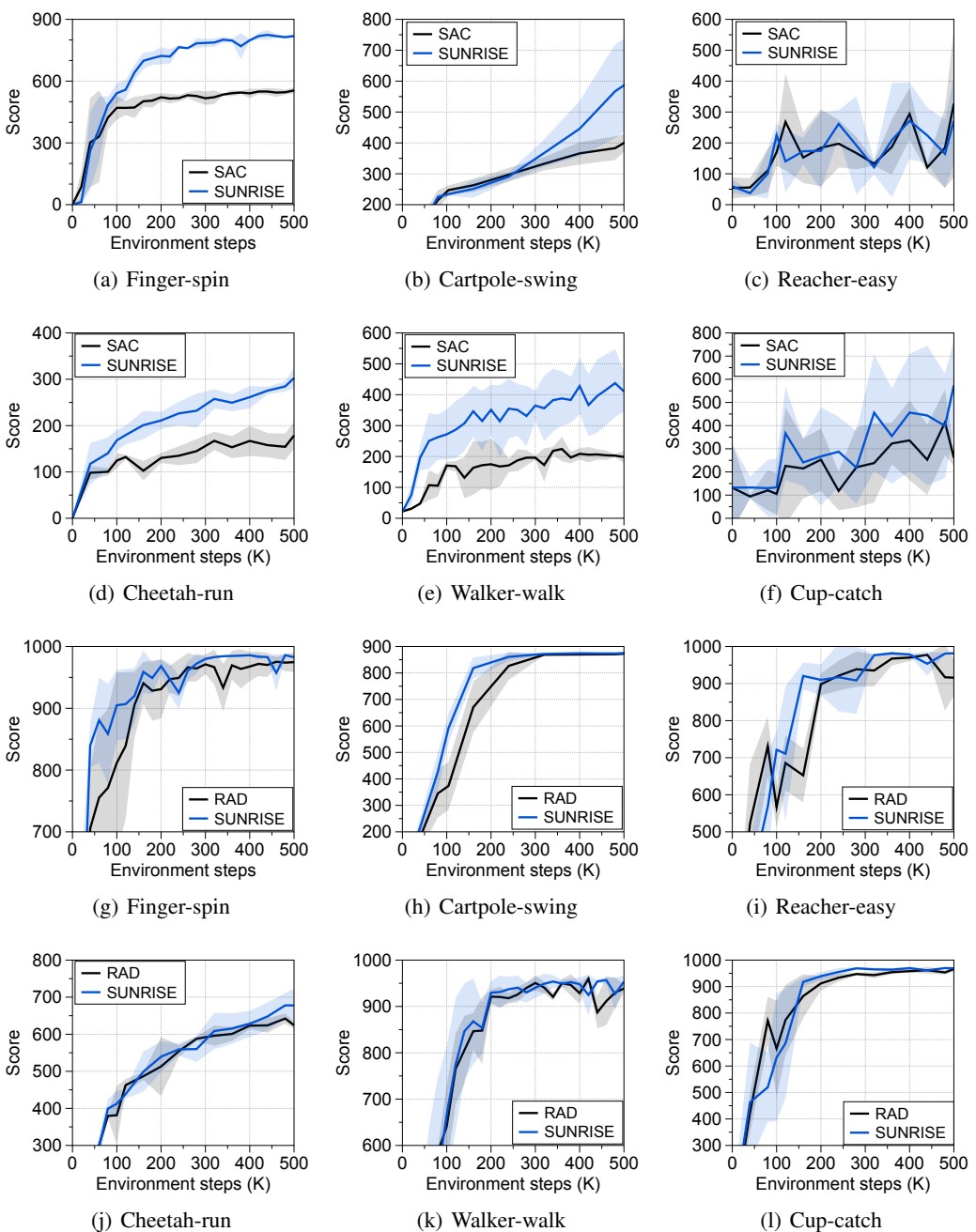

Figure 6: Learning curves of (a-f) SAC and (g-I) RAD on DeepMind Control Suite. The solid line and shaded regions represent the mean and standard deviation, respectively, across five runs.

**Learning curves**. Figure 7, Figure 8 and Figure 9 show the learning curves on all environments.

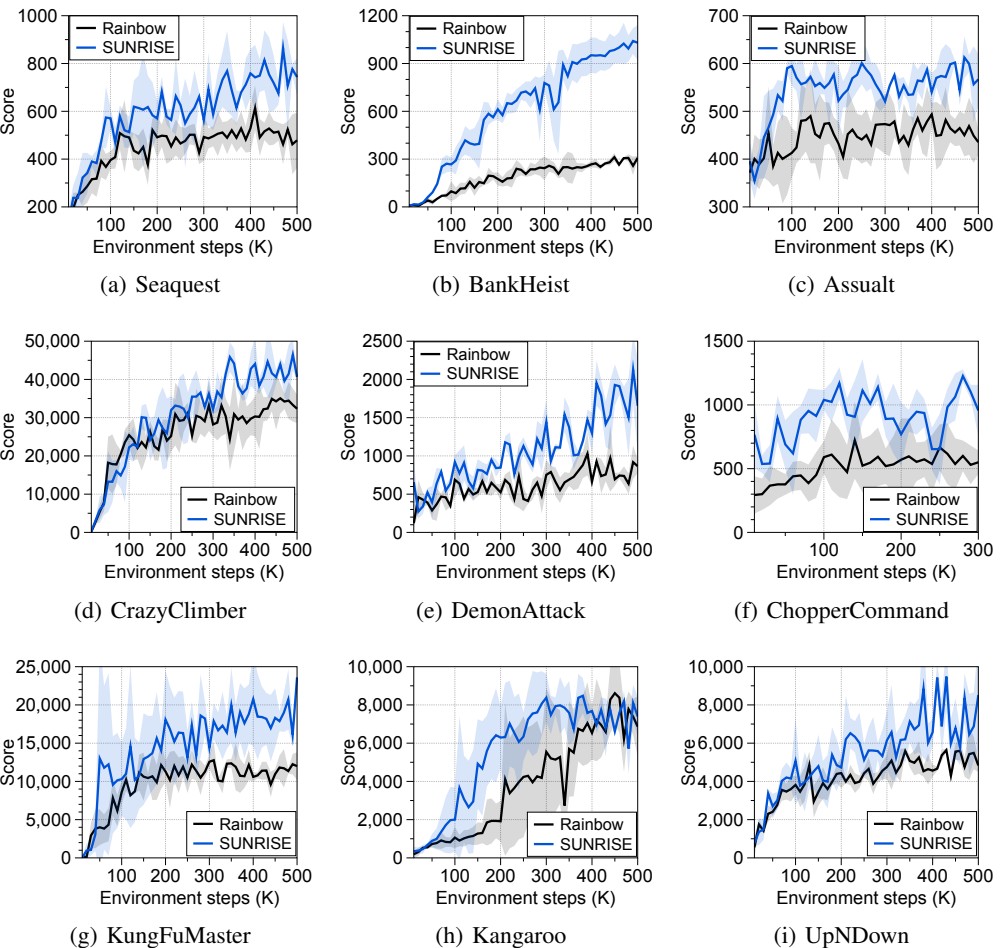

Figure 7: Learning curves on Atari games. The solid line and shaded regions represent the mean and standard deviation, respectively, across three runs.

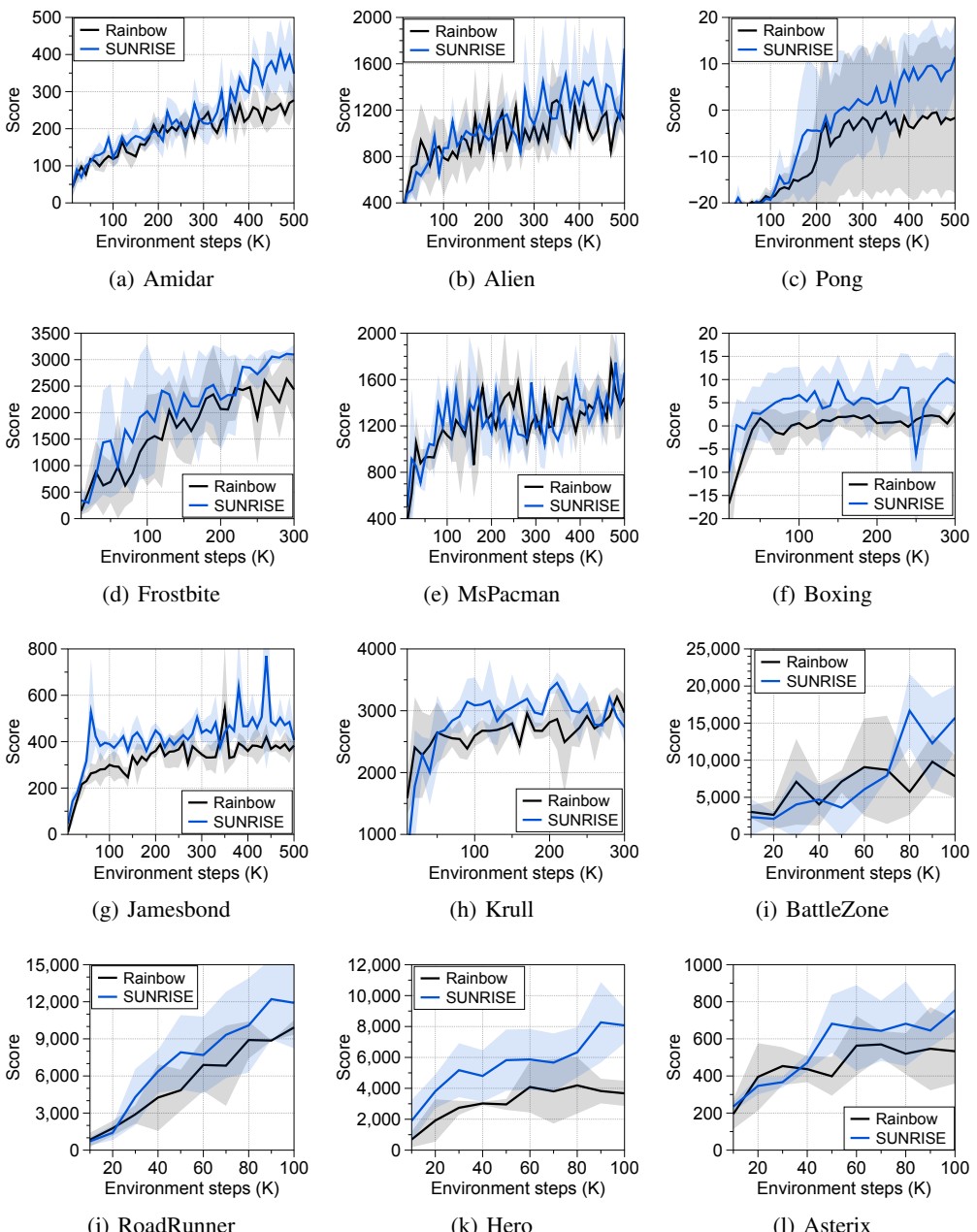

Figure 8: Learning curves on Atari games. The solid line and shaded regions represent the mean and standard deviation, respectively, across three runs.

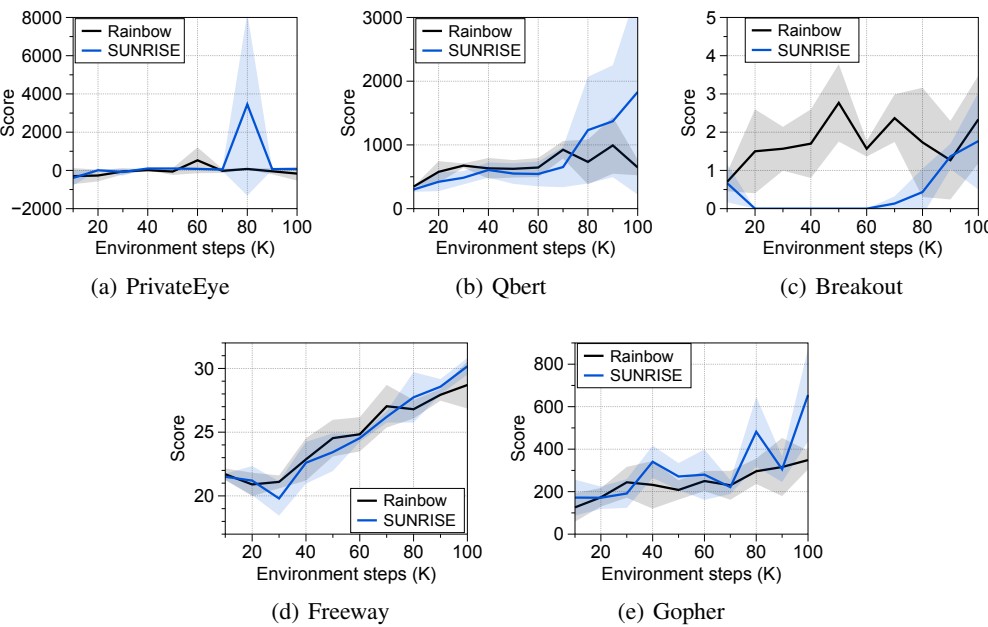

Figure 9: Learning curves on Atari games. The solid line and shaded regions represent the mean and standard deviation, respectively, across three runs.

