# OpenReview forum: "Weighted Bellman Backups for Improved Signal-to-Noise in Q-Updates"
_ICLR.cc/2021/Conference — Reject_

### Official Review · AnonReviewer1 · 2020-10-27
**An encouraging empirical result but low technical novelty and insufficient experiments to make a reliable conclusion about the role of the proprosed weighted Bellman backups**

**Rating:** 5
**Confidence:** 4

**Review:**

### Summary

This paper proposes to weight the Bellman backups according to the empirical std of Q-functions estimated by ensemble method. The paper claims that this proposed idea stabilizes and improves the learning in both continuous and discrete control tasks. It then integrates this proposed weighted Bellman backups with two of the existing advantages of ensembles: bootstrap and ucb exploration to form a unifying framework namely SUNRISE. SUNRISE is then compared with Actor-Critic and Rainbow in both discrete and continuous control tasks.

### Strong points
-	Clarity: The paper is well written and organized
-	Empirical significance: The integrated framework SUNRISE appears to have promising (yet seemly not fully convincing) performance as compared to the prior frameworks in both discrete and continuous control tasks.

### Weak points
-	Novelty: This work seems to have low novelty and low technicality. It combines several known results to integrate into a unifying framework using ensembles. The only new idea here is perhaps a specific way of reweighting the Bellman backups though the idea of reweighting the Bellman backups to stabilize the learning is already known e.g., Kumar et al. 2020.
-	Significance: In addition, at the present form I find it hard to be convinced both empirically and theoretically (or at least more elaborate explanation or intuition) why the proposed weighted Bellman backups using empirical std of Q-functions improve the signal-to-noise in Q-updates as claimed in the paper (see Questions for the authors).

###  Questions for the authors
-	In the last sentence of Section 4.1, the paper claims that “the proposed objective … has a better signal-to-noise ratio”. I would like the authors to elaborate in this claim. What is exactly considered signal and what is exactly consider noise in this context? Why down weighting the sample transitions with high variance across target Q-functions result in a better signal-to-noise ratio? Why does the weighting proposed in this paper have better signal-to-noise ratio than the weighting in Discor (Kumar et al. 2020)?
-	The paper claims that the proposed weighted Bellman updates improve signal-to-noise in Q-updates but appears to show only one experimental setting (presented in Fig. 2) where the reward is perturbed with *a standard Gaussian noise*.   For simplicity for the moment let’s call by “reward-to-noise ratio” the ratio of the magnitude of the original reward signal r(s,a) to the magnitude of the added noise. Since Section 5.3, I assume that the “reward-to-noise ratio” has something to do with the signal-to-noise ratio mentioned in the paper. Then, how the performance of the proposed weighted Bellman updates when the “reward-to-noise ratio” varies?

###  My initial recommendation
Overall, I vote for weak rejecting for the weak points mentioned above.

### My final recommendation
The authors did not fully address my points. I remain my initial score and recommend for rejection.

---

> ### Author Response · Authors · 2020-11-19
> **Response to Reviewer 1**
>
> Thank you for the helpful and valuable feedback on our paper! We have revised our draft based on your suggestions. Revised parts in the new draft are colored red. We address your comments and questions below:
>
> ---
> **Q1. Novelty**
>
> **A1.** We emphasize that our novelty is on (a) proposing a new way to reweight the Bellman backup using ensembles of Q-functions, which is more effective than DisCor [Kumar et al., 2020], and (b) showing that several techniques (with proper extensions) using ensembles are complementary and can be fruitfully integrated. As R2 and R4 pointed out, we believe that our experimental results are extensive and demonstrate the generality of the proposed method.
>
> ---
> **Q2. Theoretical justification**
>
> In DisCor [Kumar et al., 2020], they show that naive Bellman backups can suffer from slow learning in certain environments, requiring exponentially many updates. To handle this problem, they propose the weighted Bellman backups, which make steady learning progress by inducing some optimal data distribution. However, we remark that DisCor can still suffer from the error propagation issues because there is also an approximation error in estimating cumulative Bellman errors. Therefore, we consider an alternative approach that utilizes the uncertainty from ensembles. In other words, we derive more practical solutions based on their analysis and observations. Because it has been observed that the ensemble can produce well-calibrated uncertainty estimates (i.e., variance) on unseen samples, we find that ensemble-based weighted Bellman backups can give rise to more stable training and improve the data-efficiency of various off-policy RL algorithms. We added related discussions in Section 6 of the revised draft.
>
> ---
> **Q3. Signal-to-noise**
>
> **A3.** We apply the Bellman backup on the previous learned Q-function, rather than the true optimal Q-value. So, errors in the previous Q-function induce the “noise” to the learning “signal” (i.e., true Q-value) of the current Q-function. We clarified this in Section 4 of the revised draft.
>
> ---
> **Q4. More ablation on noise ratio**
>
> **A4.** Thank you for the suggestion to show our method more convincingly! We evaluate the performance of SUNRISE by adding Gaussian noise with a large standard deviation to the reward function of SlimHumanoid-ET in Figure 3(a). Even though overall performances of all methods are decreased due to noise, SUNRISE with the proposed weighted Bellman backups still achieves the best performance. We will add more experiments on other environments with a larger noise ratio by the end of the rebuttal period.
>
> ---
> Thank you for your suggestions to improve the clarity and robustness of our paper! We hope to have addressed all of your questions.

---

> > ### Comment · AnonReviewer1 · 2020-11-21
> > **Nice revision but my main concerns about novelty and empirical significance largely remain**
> >
> > Thanks for the response. I have read the response and appreciate the changes the authors have made to the new version. Overall I think this work is an interesting exploration in using ensemble for weighted Bellman updates but the idea is incremental and the experimental results are still not clear to me if the proposed idea really works:
> >
> > * Instead of directly modelling the value error as in Kumar et al 2020, this work uses ensemble for estimating the value error. This idea is incremental and straightforward. There is also not theoretical justification why this ensemble uncertainty would be better than the direct uncertainty modelling in Kumar et al 2020. Now I am not criticizing you on this nor saying that a strong technical novelty is necessary for ICLR publication, but I will put most of the weight of my evaluation in the empirical significance and applicability.
> >
> > * I still could not see if an empirical gain is actually from the ensemble uncertainty for weighted Bellman update or it is from other sources (e.g., UCB exploration based on ensemble) which are not the contribution of the paper. Since the core contribution of this paper is using ensemble uncertainty for weighted Bellman instead of direct uncertainty model as in DisCor (Kumar et al 2020), I think it is crucial to make it clear if this proposed idea is empirically better than DisCor. For example, in Fig 2 in the new version, does DisCor also have UCB exploration as SUNRISE? Do the authors have several experimental results for the comparison of just Discor and SUNRISE where both algorithms must have the same components (e.g., same exploration method and initialization techniques) except only that they use different value error estimate for weighted Bellman updates? I think the authors have tried to do that in Fig 3 but Fig 3 is only for 1 simple environment and they did not compare DisCor with SUNRISE without UCB.

---

> > > ### Author Response · Authors · 2020-11-22
> > > **Additional response to Reviewer 1**
> > >
> > > Dear Reviewer1,
> > >
> > > Thank you very much again for the additional comments.
> > >
> > > Following your suggestion, we have updated Figure 3 by adding a new variant, SUNRISE without UCB. We find that SUNRISE without UCB exploration achieves  the best performance, which clearly shows the gains from the proposed weighted Bellman backups. Due to time constraints of the rebuttal period, we only provided an additional experiment on the SlimHumanoid-ET environnement. However, in order to make it as rigorous as possible, we performed this comparison with 10 random seeds and the same amount of hyperparameter tuning for both our model and the baseline as per R3’s suggestion. Additionally,  we note that this is not a simple environment as is typical of regularly used benchmark environments.  Most existing model-based and model-free RL methods cannot solve this environment efficiently (see Table 1 in [1]) due to high-dimensional action and state spaces and environment noise. We will include this comparison on more environments in the final draft.
> > >
> > > [1] Tingwu Wang, Xuchan Bao, Ignasi Clavera, Jerrick Hoang, Yeming Wen, Eric Langlois, Shunshi Zhang, Guodong Zhang, Pieter Abbeel, and Jimmy Ba. Benchmarking model-based reinforcement learning. arXiv preprint arXiv:1907.02057, 2019.
> > >
> > > Best,
> > >
> > > Authors

---

### Official Review · AnonReviewer2 · 2020-10-28

**Rating:** 5
**Confidence:** 4

**Review:**

This submission developed an ensemble based approach to weight the Bellman backups from different agents. As claimed by the authors, the proposed method can improve the signal-to-noise ratio. To further boost the performance, the authors combined the weighted backups with a few other techniques, including UCB exploration and Bootstrap. The authors finally tested the proposed method on both continuous and discrete reinforcement learning tasks, and showed the improved or competitive performance, compared with baselines.

The proposed ensemble-based approach is interesting and the authors conducted an extensive experiments to verify its empirical performance, which I really appreciate. Compared with other baselines, the performance gap is also noticeable.

On the other hand, I am a bit concerned about whether the improvement is indeed because of the weighted backups. For example, Figure 3(a) showed that if removing UCB, the performance for SUNRISE dropped a lot. I tried to see if there are any other ablation studies w.r.t. UCB on the results in tables (removing UCB and keeping other steps the same in Algorithm 1), but did not find them. Use of UCB seems orthogonal with the weighted backup, as one is focused on exploration and the other for Q updates. Therefore, it's a bit questionable whether UCB or the proposed weighted backups is the main factor for performance improvement.

The authors claimed a few times "signal-to-noise ratio". I hope there could be more rigor here. What exactly is the definition for this term? What are the signal and noise here?

Furthermore, I also doubt about the fairness in Table 3: The results there are only for 100K interactions; however, when comparing with Figure 8, Rainbow has not become stable at 100K and the scores for some games are just too low (e.g., Breakout), compared with results in the Rainbow paper.

Could you comment on the increased complexity, when employing multiple agents? There are a few recent papers on the weighted Q updates as well, e.g.,

Song, Z., Parr, R. and Carin, L., 2019, May. Revisiting the softmax bellman operator: New benefits and new perspective. In International Conference on Machine Learning (pp. 5916-5925).

Kim, S., Asadi, K., Littman, M. and Konidaris, G., 2019, August. Deepmellow: removing the need for a target network in deep Q-learning. In Proceedings of the Twenty Eighth International Joint Conference on Artificial Intelligence.

These papers avoid the need of multiple agents and show the benefits of weighted updates, which the authors need to discuss.

---

> ### Author Response · Authors · 2020-11-19
> **Response to Reviewer 2**
>
> Thank you for the helpful and valuable feedback on our paper! We have revised our draft based on your suggestions. Revised parts in the new draft are colored red. We address your comments and questions below:
>
> ---
> **Q1. UCB exploration**
>
> **A1.** We remark that our paper verifies the effectiveness of UCB exploration for both continuous and discrete action spaces on various benchmarks, while the prior work only focused on discrete action space using Atari. Also, UCB exploration without the proposed Bellman backups can not achieve the best performance on complex environments with noisy rewards as shown in Figure 2(a).
>
> Please note that one of our contributions is to propose a unified framework combining various ensemble-based methods and show the benefits throughout extensive experiments. Because RL involves many issues (e.g. instability and exploration), RL can be improved by integrating various techniques proposed for different purposes. We hope that our work can provide such an insight to the RL community.
>
> ---
> **Q2. Definition of “signal-to-noise ratio”**
>
> **A2.** We apply the Bellman backup on the previous learned Q-function, rather than the true optimal Q-value. So, errors in the previous Q-function induce the “noise” to the learning “signal” (i.e., true Q-value) of the current Q-function. We clarified this in Section 4 of the revised draft.
>
> ---
> **Q3. Computation**
>
> **A3.** When we have $N$ ensemble agents, our method requires $N\times$ inferences for weighted Bellman backups and $2N\times$ inferences ($N$ for actors and $N$ for critics). However, we remark that our method can be more computationally efficient because it is parallelizable. Also, as shown in Figure 3(c) of the revised draft, the gains from SUNRISE can not be achieved by simply increasing the number of updates/parameters. We added related discussions in Section 6 of the revised draft.
>
> ---
> **Q4. Related works**
>
> **A4.** We cited relevant works on the weighted Q updates in Section 2. Thank you very much for your suggestion.
>
> ---
> **Q5. Table 3**
>
> **A5.** We focus on performance after 100K interactions because of the recent emphasis on making RL more sample efficient (e.g. recent papers: CURL [Srinivas et al., 2020] and SimPLe [Kaiser et al., 2020]). However, we agree it is important to also verify the effectiveness of our method at large samples. So, we report the learning curves with increased number of samples in Figure 7-9, where the gain from SUNRISE becomes even more significant when training longer.
>
> ---
> Thank you for your suggestions to improve the clarity and robustness of our paper! We hope to have addressed all of your questions.

---

> > ### Comment · AnonReviewer2 · 2020-11-21
> > **Helpful response but the main concern is still there**
> >
> > Thank you for the response, which indeed addressed some of my concerns. I still think this paper has an interesting motivation; however, the contribution seems a bit incremental and the evaluation is not quite rigorous.
> >
> > As stated in my original comments, SUNRISE combined the weighted backups with UCB exploration. It's fine to combine with UCB, but at least there should be some evidence to show that using the proposed weighted backups only works, otherwise it's unclear which one contributed more. This concern was also raised by Reviewer 1, and I agree with the comments there.
> >
> > Regarding Table 3, thanks for providing more results in Figures 7-9. I am a bit worried as the scores there are much worse than Figure 5 in the original Rainbow paper(https://arxiv.org/pdf/1710.02298.pdf). I understand that it may be due to different implementations, but even for some simple games such as Pong, the scores are much lower than expected.

---

> > > ### Author Response · Authors · 2020-11-22
> > > **Additional response to Reviewer 2**
> > >
> > > Dear Reviewer2,
> > >
> > > Thank you very much again for the additional comments.
> > >
> > > ---
> > > Following your suggestion, we have updated Figure 3 by adding a new variant, SUNRISE without UCB. We find that SUNRISE without UCB exploration achieves  the best performance, which clearly shows the gains from the proposed weighted Bellman backups. Due to time constraints of the rebuttal period, we only provided an additional experiment on the SlimHumanoid-ET environnement. However, in order to make it as rigorous as possible, we performed this comparison with 10 random seeds and the same amount of hyperparameter tuning for both our model and the baseline as per R3’s suggestion. Additionally,  we note that this is not a simple environment as is typical of regularly used benchmark environments.  Most existing model-based and model-free RL methods cannot solve this environment efficiently (see Table 1 in [1]) due to high-dimensional action and state spaces and environment noise. We will include this comparison on more environments in the final draft.
> > >
> > > ---
> > > We also remark that data-efficient Rainbow [2] uses different hyper-parameters and encoder architecture. Because of that, compared to the original Rainbow, the scores are lower on some environments. Again, we’d like to emphasize that we used this benchmark because of the recent emphasis on making RL more sample efficient (e.g. recent papers: Data-efficient Rainbow [2], CURL [3], SimPLe [4], and DrQ[5]). However, we also agree it is important to also verify the effectiveness of our method at standard setup (based on our observations when learning for longer, we expect that the gains of our method would be more significant). Due to time constraints of the rebuttal period, we will add more experiments using the original Rainbow in the final version.
> > >
> > > ---
> > > [1] Tingwu Wang, Xuchan Bao, Ignasi Clavera, Jerrick Hoang, Yeming Wen, Eric Langlois, Shunshi Zhang, Guodong Zhang, Pieter Abbeel, and Jimmy Ba. Benchmarking model-based reinforcement learning. arXiv preprint arXiv:1907.02057, 2019.
> > >
> > > [2] Hado P van Hasselt, Matteo Hessel, and John Aslanides. When to use parametric models in reinforcement learning? In Advances in Neural Information Processing Systems, 2019.
> > >
> > > [3] Aravind Srinivas, Michael Laskin, and Pieter Abbeel. Curl: Contrastive unsupervised representations for reinforcement learning. In International Conference on Machine Learning, 2020.
> > >
> > > [4] Lukasz Kaiser, Mohammad Babaeizadeh, Piotr Milos, Blazej Osinski, Roy H Campbell, Konrad Czechowski, Dumitru Erhan, Chelsea Finn, Piotr Kozakowski, Sergey Levine, et al. Model-based reinforcement learning for atari. In International Conference on Learning Representations, 2020.
> > >
> > > [5] Kostrikov, I., Yarats, D. and Fergus, R., Image augmentation is all you need: Regularizing deep reinforcement learning from pixels. arXiv preprint arXiv:2004.13649, 2020.
> > >
> > > Best,
> > >
> > > Authors

---

### Official Review · AnonReviewer4 · 2020-10-28
**The approach proposed in the paper is interesting and the results suggest that it is successfully able to outperform current state-of-the art approaches in several benchmark domains from the literature.**

**Rating:** 8
**Confidence:** 4

**Review:**

= Overview =

The paper proposes SUNRISE, an approach to reinforcement learning that leverages ensembles of agents to build more robust RL updates. SUNRISE comprises a number of similar agents  (in the paper, SAC agents) that perform parallel updates. Sample transitions for which there is larger variability (across the ensemble) in the estimates of the next-step Q-values are down-weighted in the computation of the loss, thus potentially rendering the learned Q-function more robust to noise.

The proposed approach is then combined with bootstrapping masks and UCB exploration, and is shown to outperform a number of state-of-the-art approaches in several benchmark domains from the RL literature.

= Positive points =

The paper is clearly written. Additionally, the proposed approach is sensible and the empirical evaluation is, in my perspective, quite comprehensive: SUNRISE is evaluated in a broad collection of domains in the RL literature.

= Negative points =

The paper would benefit, in my opinion, from additional discussion regarding: (a) the impact of the use of bootstrap with random initialization; and (b) the computational complexity of SUNRISE (even if the paper does briefly discuss the latter in Section 5.2)

= Comments =

I quite enjoyed reading the paper. The problem addressed is a relevant problem in RL, and the approach proposed in the paper is, in my opinion, simultaneously simple and sensible. The paper provides a solid empirical evaluation, covering a broad range of domains and comparing with multiple state of the art approaches from the literature. The results show that SUNRISE compares favorably -- in terms of performance -- with several of these other methods in multiple domains.

There are, however, two aspects that I would like to see discussed at greater length. On one hand, the paper proposes the use of bootstrapping masks and random initialization to induce variety in the ensemble. While the paper introduces both bootstrapping and UCB exploration as a "useful complement", it seems to me that this is quite central to the performance of the algorithm. Is this correct? In fact, without this device, the agents in the ensemble would essentially train from the same replay buffer, so variability would only come from the initialization. It is a pity that this particular element isn't included in the ablation study, for I would like to gain a clearer understanding on how critical this device is for the performance of the algorithm.

One other aspect that I would like to see discussed is regarding the computational complexity of the proposed approach. The paper remarks that SUNRISE is more computationally efficient than competing methods such as POPLIN and PETS, and being an ensemble method, I expect it to be naturally heavier than non-ensemble approaches such as standard SAC. However, I would like to understand how much more computation such a method involves. In particular the computation of the Bellman weights requires multiple passes through the critic network, as does the UCB exploration policy, and I was wondering how much more computation this entails.

In spite of the above aspects, I again remark that I quite enjoyed the paper.

---

> ### Author Response · Authors · 2020-11-19
> **Response to Reviewer 4**
>
> Thank you for the helpful and valuable feedback on our paper! We have revised our draft based on your suggestions. Revised parts in the new draft are colored red. We address your comments and questions below:
>
> ---
> **Q1. Impact of bootstrap**
>
> **A1.** Thank you for your pointer. As R1 and you mention, we agree that Bootstrap with random initialization is a necessary component for uncertainty estimation and we can apply the proposed weighted Bellman backups and UCB exploration based on that. We clarified this in the abstract, introduction and Section 4. Also, by following your suggestion, we plan to include the ablation on Bootstrap by the end of the rebuttal period.
>
> ---
> **Q2. Computation complexity**
>
> **A2.** When we have $N$ ensemble agents, our method requires $N\times$ inferences for weighted Bellman backups and $2N\times$ inferences ($N$ for actors and $N$ for critics). However, we remark that our method can be more computationally efficient because it is parallelizable. Also, as shown in Figure 3(c) of the revised draft, the gains from SUNRISE can not be achieved by simply increasing the number of updates/parameters. We added related discussions in Section 6 of the revised draft.
>
> ---
> Thank you for your suggestions to improve the clarity and robustness of our paper! We hope to have addressed all of your questions.

---

> > ### Comment · AnonReviewer4 · 2020-11-21
> > **Response to authors**
> >
> > I would like to thank the authors for the clarifications. I am happy with their response.

---

### Official Review · AnonReviewer3 · 2020-10-29

**Rating:** 3
**Confidence:** 4

**Review:**

This paper proposes to use uncertainty estimates from an ensemble of action-values, to provide a weighting on the updates in Q-learning. The main idea is to use the sigmoid of the negative of this uncertainty in the next state, to produce a weighting between 0.5 and 1 to downweight updates with high uncertainty targets. This uncertainty estimate from the ensemble is also used to improve exploration, in a combined algorithm called Sunrise that leverages learning an ensemble in these two ways.

The idea of using weighted Bellman updates is, as far as I know, novel. The evidence for the idea, however, needs more work. First, the weighted update in Eq (4) is not motivated from first principles. Second, the empirical evidence is weak because the experiments highlighting the role of the weighting do not demonstrate significant differences.

The first issue is the justification for the approach. The ensemble of Q-learning agents is trained using the weighting, derived from that ensemble. There are natural questions as to the interaction between the ensemble uncertainty estimates and the ensemble estimates. Does it result in any instability? What is the final point of convergence? Does it change the solution?

But, one could argue that that is not much of a problem, since the weighting w(s,a) is always between 0.5 and 1, so it is not that skewed. Then the question arises how much it is helping, and why this small reduction in weight helps. This is particularly important to ask, considering the algorithm requires an ensemble to be learned, with subsets of data used for each action-value. There is a lot of effort expended for that weighting.

The experiments then do include ablations, to examine the effect of these weightings. Unfortunately, the results are inconclusive. The experimental time spent must have been large to get all the results in this paper, across so many environments and algorithms. But, the ablations themselves are not sufficiently in-depth to provide insight into the idea and algorithm. The results in Figure 2 are key, since that figure examines Sunrise with and without the weighting. Due to the variance across runs, with only 4 runs, there are large standard errors (and so even larger 95% confidence intervals); it is hard to conclude that weighting is helping. The additional results in Figure 5 in the appendix have a similar issue.

The results in Figure 3, which motivate the exploration utility, are more clear in Cartpole. This provides some motivation for learning ensembles, so they can be used for exploration. But, this exploration approach with ensembles is an existing method. The main novelty in this work is the weighting.

I highly recommend taking a few domains and carefully studying the impact of the weight. More runs would help for significance, as well as parameter sensitivity analysis to gain insight into the generality of the improvement. Sometimes performance gains are from hyperparameter tuning, rather than from the utility of an idea; here, you really want to know if and why this weighting improves performance.

As a more minor comment, Sunrise is pitched as combining three ideas for using ensembles: your weighting, bootstrapping and UCB exploration. However, I see Sunrise as combining two ideas: weighting and UCB exploration. The Bootstrap DQN approach gives you a way to learn your ensemble of bootstrap models, so that it provides a useful uncertainty estimate. Given that ensemble, you can then use it to compute a weighting and optimistic action. It would be more clear to separate it out that way, rather than saying "Furthermore, since our weighted Bellman backups rely on maintaining an ensemble, we investigate how weighted Bellman backups interact with other benefits previously derived from ensembles: (a) Bootstrap; (b) UCB Exploration." The bootstrap is arguably not a benefit, but an approach to obtain confidence (uncertainty estimates).

Minor comments:
1. Bootstrap DQN is listed under "Ensemble Methods in RL", rather than under "Exploration in RL", but is it an exploration approach.
2. "Recently, Kumar et al. (2020) showed that this error propagation can cause inconsistency and unstable convergence." The terms inconsistency and unstable convergence should be explained, since they seem like technical terms.
3. Bellman backup seems to be used to describe the squared error to the expectation over next action, in Equation (2), and then to a stochastic sample of the action in (4). Which is it?
4. What is meant by the signal-to-noise in Q-updates?
5. A natural baseline to include is to tune an agent that uses random weights between 0.5 and 1 in the update, but keeping other parts of Sunrise the same. The ablation removes the weighting all together, which is also important to include. But, it's worthwhile observing if random weights performs similarly, especially if that agent is tuned.

------------ Update
Thank you for the clear reply. Unfortunately, I remain concerned about the significance of experiments. I mentioned above that 4 or 5 runs is typically not enough, and because the standard errors are overlapping, the differences could be due to chance. The addition of a result with 10 runs is a good step. But, as part of the reply, the authors state: "Figure 3(a) shows the learning curves of all methods on the SlimHumanoid-ET environment over 10 random seeds. First, one can not that SUNRISE with random weights (red curve) is worse than SUNRISE with the proposed weighted Bellman backups (blue curve). Additionally, even without UCB exploration, SUNRISE with the proposed weighted Bellman backups (purple curve) outperforms all baselines. This implies that the proposed weighted Bellman backups can handle the error propagation effectively even though there is a large noise in reward function." However, if you look at this figure, the error bars all still overlap. 10 random seeds is still not enough.

I am also not confident that the issue will be remedied, as the authors additionally state in the rebuttal: "we believe that SUNRISE is evaluated in a broad collection of domains in the RL literature and the performance gap is also noticeable." An insignificant gap across many domains does not tell us anything. Actually, if you take the runs and tried to do significance tests by pooling all the runs across environments, then maybe the result might actually be significant. But, of course, there will be higher variance due to differences in the environments, so it is not obvious this would be true. Nonetheless, this could be a natural next step.

---

> ### Author Response · Authors · 2020-11-19
> **Response to Reviewer 3**
>
> Thank you for the helpful and valuable feedback on our paper! We have revised our draft based on your suggestions. Revised parts in the new draft are colored red. We address your comments and questions below:
>
> ---
> **Q1. The theoretical justification for the proposed weighted Bellman backups**
>
> **A1.** In DisCor [Kumar et al., 2020], they show that naive Bellman backups can suffer from slow learning in certain environments, requiring exponentially many updates. To handle this problem, they propose the weighted Bellman backups, which make steady learning progress by inducing some optimal data distribution. However, we remark that DisCor can still suffer from the error propagation issues because there is also an approximation error in estimating cumulative Bellman errors. Therefore, we consider an alternative approach that utilizes the uncertainty from ensembles. In other words, we derive more practical solutions based on their analysis and observations. Because it has been observed that the ensemble can produce well-calibrated uncertainty estimates (i.e., variance) on unseen samples, we find that ensemble-based weighted Bellman backups can give rise to more stable training and improve the data-efficiency of various off-policy RL algorithms. We added related discussions in Section 6 of the revised draft.
>
> ---
> **Q2. Justification for the confidence weight in eq (4)**
>
> **A2.** Thank you for the suggestion to show our method more convincingly! We compare with another variant of SUNRISE, which updates Q-functions with random weights sampled from $[0.5, 1.0]$ uniformly at random on SlimHumanoid-ET environment with noisy reward. We report the performance over 10 random seeds in Figure 3(a), where SUNRISE with the proposed weighted Bellman backups outperforms all baselines including SUNRISE with random weight and only with UCB exploration. This implies that the proposed weighted Bellman backups can handle the error propagation effectively even though there is a large noise in the reward function.
>
> ---
> **Q3. UCB exploration**
>
> **A3.** We remark that our paper verifies the effectiveness of UCB exploration for both continuous and discrete action spaces on various benchmarks, while the prior work only focused on discrete action space using Atari. Also, UCB exploration without the proposed Bellman backups can not achieve the best performance on complex environments with noisy rewards as shown in Figure 2(a).
>
> Please note that one of our contributions is to propose a unified framework combining various ensemble-based methods and show the benefits throughout extensive experiments. Because RL involves many issues (e.g. instability and exploration), RL can be improved by integrating various techniques proposed for different purposes. We hope that our work can provide such an insight to the RL community.
>
> ---
> **Q4. Definition of Signal-to-noise**
>
> **A4.** We apply the Bellman backup on the previous learned Q-function, rather than the true optimal Q-value. So, errors in the previous Q-function induce the “noise” to the learning “signal” (i.e., true Q-value) of the current Q-function. We clarified this in Section 4 of the revised draft.
>
> ---
> **Q5. BootstrapDQN in Related work**
>
> **A5.** We updated the related work accordingly. Thank you very much for your suggestion.
>
> ---
> **Q6. Equation 2 & 4**
>
> **A6.** By following SAC, we approximate the expectation using a stochastic sample of the action in (4). You can interpret this one as the Monte Carlo approximation (with one sample).
>
> ----
> **Q7. Bootstrap**
>
> **A7.** Thank you for your pointer. As R1 and you mention, we agree that Bootstrap with random initialization is a necessary component for uncertainty estimation and we can apply the proposed weighted Bellman backups and UCB exploration based on that. We clarified this in the abstract, introduction and Section 4.
>
> ---
> Thank you for your suggestions to improve the clarity and robustness of our paper! We hope to have addressed all of your questions.

---

> > ### Comment · AnonReviewer3 · 2020-11-23
> > **R3 Response to Author Response**
> >
> > Thank you for the in-depth response. Unfortunately, a primary concern remains unaddressed, which is the lack of significance in the results. With just 4 or 5 runs, the standard errors overlap significantly, and results could be due to chance. This might be a confusing comment, given that so much of the community asks for bigger agents on more environments, at the sacrifice of more runs and careful hyperparameter selection. But, that is not a sound empirical methodology. Issues with a small number of runs have been clearly highlighted in: "Deep Reinforcement Learning that Matters", Henderson et al. 2018.
> >
> > Additionally, you state: "The optimal parameters are chosen to achieve the best performance on training environments." This means the other agents are not tuned, but your agents gets more chance to do well by picking the best hyperparameters. This tuning might explain performance differences. More hyperparameters should not confer an advantage, and you should attempt to sweep a similar number of parameters for the other agents. It's also not clear here if you meant that the best hyperparameters per environment were chosen (or across environments) and if you then reported the same runs for the best hyperparameters or if you took that best hyperparameter setting and then ran the agent again (to get an independent set of runs).
> >
> > The response also does not provide clarity on the main first concern: how does this reweighting affect the solution? In an idealized setting, would you expect it to eventually go to 1? A proof is not needed, but a discussion on the potential impacts of this choice on solution quality are important. Maybe here you are saying DisCor already provides the theoretical discussion needed to motivated weighted Bellman backups, and then you are simply providing a more practical alternative. If so, it even more suggests that you should start with a more clear explanation of DisCor, what is known there, and why your idea is needed beyond their approach.

---

> > > ### Author Response · Authors · 2020-11-23
> > > **Additional response to R3**
> > >
> > > Dear Reviewer3,
> > >
> > > Thank you very much again for the additional comments.
> > >
> > > ---
> > > **Q1.** Random seeds
> > >
> > > **A1.** For additional experiments on the SlimHumanoid-ET environnement with large environment noise, we report the performance over **10** random seeds (please see Figure 3(a) in the revised draft), where SUNRISE with the proposed weighted Bellman backups has  better performance compared to the baselines. Due to time constraints of the rebuttal period, we only provided an additional experiment on the SlimHumanoid-ET environnement but we will increase the number of random seeds in the final version.
> > >
> > > As other reviewers (R1, R2, R4) mention, we believe that SUNRISE is evaluated in a broad collection of domains in the RL literature and the performance gap is also noticeable.
> > >
> > >
> > > ---
> > > **Q2.** Hyper-parameters
> > >
> > > **A2.** For our method, the best hyperparameters were chosen for each environment with the exception of Atari where it is common to use the same hyperparameters for all tasks in the benchmark. However, by following the same strategy, we also did hyper-parameter tuning for baselines (such as SAC and DisCor) in Figure 2&3. Also, for some baseline methods (CURL, SimPLe, DrQ and SLAC) in Table 1,2 &3, we report the best numbers reported in prior works, where optimal parameters are chosen to achieve the best performance on training environments. So, we believe that our agents are evaluated on the fair condition and do not get more benefit from hyper-parameter tuning.
> > >
> > > ---
> > > **Q3.** Justification
> > >
> > > **A3.** If uncertainty estimates from ensembles can approximate the cumulative Bellman errors, the proposed weighted Bellman backups can make Q-function converge to optimal Q-function more quickly (exponential to polynomial based on observations from DisCor). In an ideal case (where there is no approximation in the target Q-function), weights go to 1 eventually. As you pointed out, we provide a more practical alternative for weighted Bellman backups and evaluate the effectiveness of it throughout extensive experiments. Additionally, by following R1 and R2’ suggestions, we’ve provided some evidence in Figure 3 of the revised draft.
> > >
> > > Best,
> > >
> > > Authors

---

### Decision · Program_Chairs · 2021-01-07
**Final Decision**

**Decision:**

Reject

**Comment:**

The paper is acknowledged by all the reviewers as making a novel contribution -- the proposal to reweight state-action pairs depending on the variation in their Q-value estimates during learning. However, despite its extensive reporting of numerical experiments, its arguments in favor of the proposed approach are found to be wanting on both empirical and theoretical fronts. Reviewer 3 points out (correctly, in my opinion) that 5 (or even 10 in the updated version) independent trials are not sufficient to establish the validity of the approach up to statistical significance, and that even a well-reasoned heuristic explanation of why reweighting is expected to work in terms of reducing Q-value estimation error is missing. I agree with this point, which also struck me when reading the submission myself, that at the very least, the submission ought to contain a basic (and not necessarily rigorous) argument as to why the variance reduction ostensibly achieved due to reweighting should lead the estimation algorithm to the right Q-function in a general function approximation setting. For instance, even in the simplest multi-armed bandit setting, it is of interest to ask why this procedure should perform consistently without introducing unwanted bias in an unforeseen sense, and a clear explanation offered for this would be interesting. Another important concern that most reviewers are left with is about the lack of sufficient insight into the action of the UCB mechanism against the backdrop of the reweighting procedure (reviewers 1, 2, 3). I hope that the author(s) assimilate the feedback to strengthen the paper's main pitch further and make a strong case in the near future.